# DREAMBENCH++: A HUMAN-ALIGNED BENCHMARK FOR PERSONALIZED IMAGE GENERATION

**Yuang Peng**[1,4,†]    **Yuxin Cui**[1,†]    **Haomiao Tang**[1,†]    **Zekun Qi**[1]    **Runpei Dong**[2,¶]
**Jing Bai**[3]    **Chunrui Han**[4,‡]    **Zheng Ge**[4]    **Xiangyu Zhang**[4]    **Shu-Tao Xia**[1,¶]

[1]Tsinghua University    [2]UIUC    [3]UCAS    [4]StepFun

**Project Page:  DreamBench++**

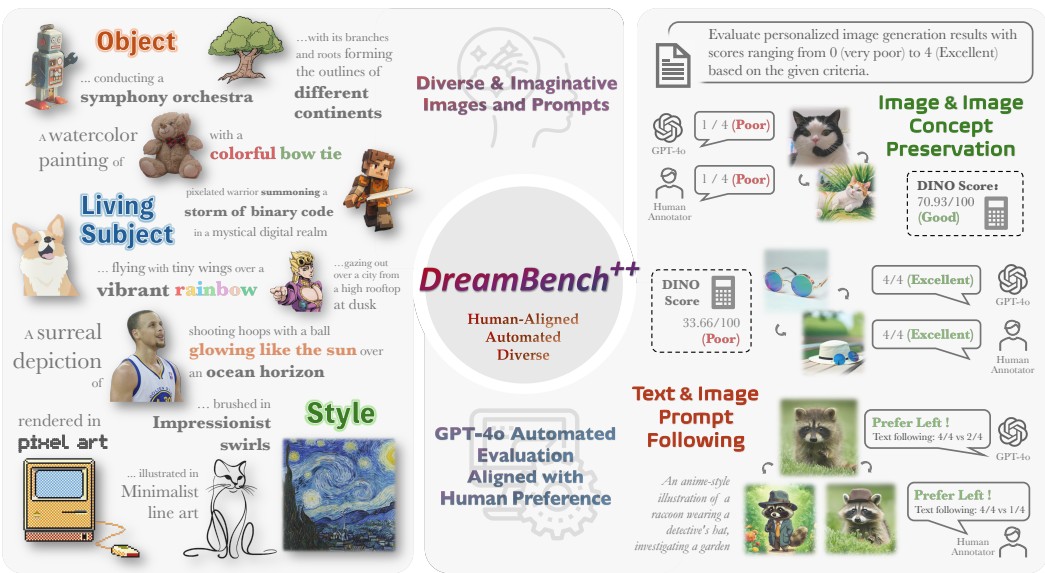

Figure 1: **Overview of DREAMBENCH++.** We collect diverse images and prompts as the base dataset and utilize GPT-4o for *automated* evaluation *aligned with human preference*. To evaluate personalized image generation, DREAMBENCH++ mainly focuses on image & image *concept preservation* and text & image *prompt following* assessments.

## ABSTRACT

Personalized image generation holds great promise in assisting humans in everyday work and life due to its impressive ability to creatively generate personalized content across various contexts. However, current evaluations either are automated but misalign with humans or require human evaluations that are time-consuming and expensive. In this work, we present DREAMBENCH++, a human-aligned benchmark that advanced multimodal GPT models automate. Specifically, we systematically design the prompts to let GPT be both human-aligned and self-aligned, empowered with task reinforcement. Further, we construct a comprehensive dataset comprising diverse images and prompts. By benchmarking 7 modern generative models, we demonstrate that DREAMBENCH++ results in significantly more human-aligned evaluation, helping boost the community with innovative findings.

---

[†]Equal contribution. [‡]Project leader. [¶]Corresponding authors.

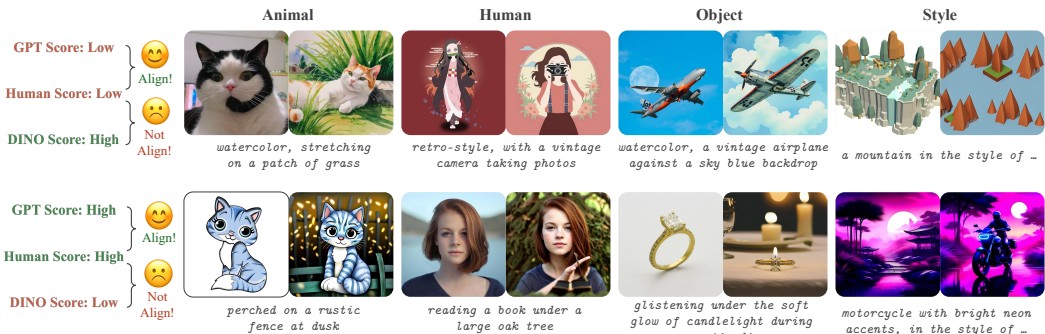

Figure 2: **Qualitative comparison of concept preservation evaluation** between DREAM-BENCH++ and traditional DINO (Caron et al., 2021). DINO often fails to yield human-aligned evaluation while our DREAMBENCH++ succeeds using multimodal GPT models as the evaluator.

# 1 INTRODUCTION

Driven by the significant advances in large-scale text-to-image (T2I) generative models (Rombach et al., 2022; Ramesh et al., 2021; Betker et al., 2023; Ramesh et al., 2022; Nichol et al., 2022; Saharia et al., 2022b; Yu et al., 2022; Chang et al., 2023; Gafni et al., 2022; Ding et al., 2021; 2022; Balaji et al., 2022; Kang et al., 2023; Dong et al., 2024), it is now possible to generate images conditioned on not only arbitrary text prompts but also by given reference images—*personalized* image generation (Ruiz et al., 2023; Gal et al., 2023a; Li et al., 2023a; Ye et al., 2023; Kumari et al., 2023; Gal et al., 2023b; Arar et al., 2023; Chen et al., 2023c; Jia et al., 2023; Chen et al., 2023a; Xiao et al., 2023; Tewel et al., 2023; Wei et al., 2023; Ma et al., 2023; Hua et al., 2023; Wang et al., 2024b; Lv et al., 2024; Wang et al., 2024a; Chen et al., 2023b; Tumanyan et al., 2023; Zhou et al., 2024; Tan et al., 2024; He et al., 2024c; Wang et al., 2024c; Wu et al., 2024a; He et al., 2024a; Xiao et al., 2024; Arar et al., 2024; Huang et al., 2024b;a; Pang et al., 2024a;b; Qiu et al., 2023; Hu et al., 2024). In general, to be useful as an artistic creation tool for inspiration or products (Yacoubian, 2022; Xie et al., 2024), the following two basic criteria must be fulfilled: **i) Prompt following** (image & prompt consistency). Generated images must follow the prompt description, which is a requirement shared with vanilla T2I generation (Betker et al., 2023; Ramesh et al., 2022). **ii) Concept preservation** (image & image consistency). For personalized image generation, the concept of the reference image, *i.e.*, the main subject's semantic details (*e.g.*, facial characters) or high-level abstractions (*e.g.*, overall style), must be preserved. For example, a user may want to "imagine his own dog traveling around the world" (Ruiz et al., 2023), and the generated dog must be the same as his but traveling.

To meet the aforementioned requirements, numerous efforts have been devoted. One line of fine-tuning-based works focuses on fine-tuning general T2I models to specialist personalization models by reproducing specific concepts present in training sets (Ruiz et al., 2023; Gal et al., 2023a; Chen et al., 2023c; Kumari et al., 2023). Meanwhile, another line of encoder-based works, instead, achieves concept-preservation by training features adaptation to inject reference image features into a general T2I model (Ye et al., 2023; Gal et al., 2023b; Arar et al., 2023; Dong et al., 2024; Sun et al., 2024a;b; Pan et al., 2024). Despite remarkable progress, one question arises: *can we comprehensively evaluate these models to figure out which technical route is superior and where to head?*

In this work, we aim to answer this question by developing a new benchmark that properly evaluates personalized T2I models driven by the above two requirements. We present DREAMBENCH++, a comprehensive benchmark designed based on the following *de-facto* principled advantages:

*1.* **Human-Aligned**  As shown in Fig. 2, traditional metrics like DINO (Caron et al., 2021) and CLIP (Radford et al., 2021) often result in significant discrepancies from humans. This is caused by the image similarity measurement nature of DINO and CLIP models, and thus crowd-sourced *human evaluation* is typically necessary for obtaining a correct *quantitative* understanding of generated images (Lee et al., 2023; Ku et al., 2024; Xu et al., 2023). Therefore, different from existing works that utilize CLIP and DINO as metrics that may be humanly misaligned, our DREAMBENCH++ demonstrates surprisingly consistent evaluation results aligned with humans. For instance, by evaluating 7 modern models, DREAMBENCH++ achieves **79.64%** and **93.18%**

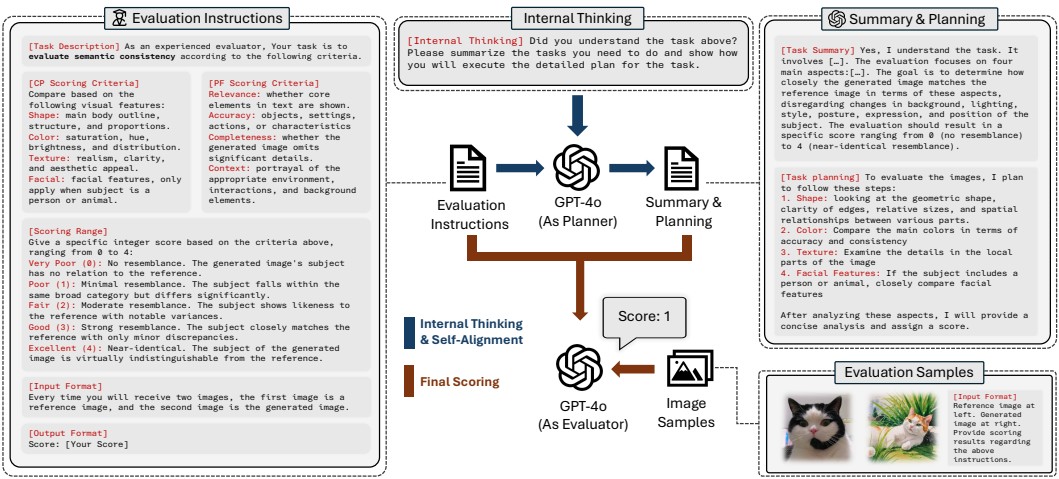

Figure 3: **Overall procedure of prompting GPT-4o for automated evaluation.** Human-written meta-prompts provide task description, scoring criteria, range, and format. GPT-4o then follows reasoning instructions for self-aligned task summarization and planning. Finally, all prompts and reasoning outputs are combined with image samples for score generation.

agreement with human's evaluation in concept preservation and prompt following capabilities, respectively. Notably, it is **+32.59%** and **+37.23%** higher than traditional DINO and CLIP metrics.

2. **Automated** However, it is non-standardized and expensive to perform high-quality human evaluations. To address this challenge, DREAMBENCH++ achieves automated but human-aligned evaluation by using advanced multimodal GPT models such as GPT-4o (OpenAI, 2024) as metrics. The challenges lie in two aspects: i) prompt design and ii) reasoning procedure for scoring. We systematically standardize the automated GPT evaluation by first designing the *evaluation instructions* that provide overall task requirements, where language is a general interface for instructing human preference. Inspired by Self-Align (Sun et al., 2023), we instruct GPT to conduct *internal thinking* that aligns itself for better task and preference understanding. Then, GPT provides the *summary & planning* for the task and scoring criteria, and the final scores are provided with optional *chain-of-thought (CoT)* (Wei et al., 2022; Zhang et al., 2023d).

3. **Diverse** To avoid bias from low-diversity evaluation data, DREAMBENCH++ compiles a wide range of images, covering varying levels of difficulty from simpler animals and styles to more complex human subjects, objects, and non-natural styles (see Fig. 1). Unlike DreamBench (Ruiz et al., 2023), which includes only 30 subjects and 25 prompts, DREAMBENCH++ significantly expands the dataset to 150 images and 1,350 prompts—**5×** and **54×** more, respectively. While CustomConcept101 (Kumari et al., 2023) offers 101 subjects, its diversity is limited by repetitive image categories and a focus on photorealistic styles, with simple prompts that restrict its ability to evaluate models on more complex tasks. Consequently, DREAMBENCH++ enables more robust and comprehensive conclusions in model evaluation.

**Takeaways** We present some insightful findings from evaluating seven modern personalized T2I models: i) DINO-based ratings prioritize overall shape and color over detailed features, making them suboptimal for evaluating personalized image generation; ii) The primary goal is to achieve a Pareto optimal balance between concept preservation and prompt adherence. Among the models, Dream-Booth (Ruiz et al., 2023) excels in preserving detailed visual features while closely following text prompts; iii) Current models perform well in animal and style categories but struggle with human images due to sensitivity to facial details and diverse object categories. While existing work (Wang et al., 2024b; Valevski et al., 2023; Yan et al., 2023; Ye et al., 2023; Xiao et al., 2023) addresses facial feature preservation, the challenge of object diversity remains underexplored.

We are presenting DREAMBENCH++ with open-sourced codes and evaluation standardization to promote innovation within the research community. In addition, we believe our design of the human-aligned & automated evaluation using advanced foundation models is robust and transferrable to other domains and foundation models (*e.g.*, GPT-5 in the future).

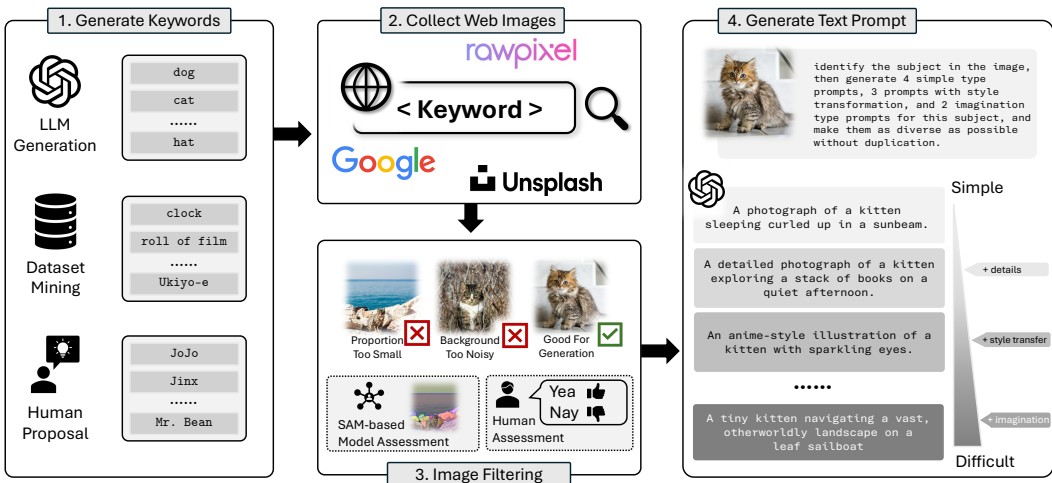

Figure 4: **Dataset construction process of our DREAMBENCH++.** We generate keywords using GPT-4, existing datasets, and human input, then collect corresponding web images. After filtering out low-quality images via model and human assessment, the high-quality ones are used as input for GPT-4o to generate text prompts of varying difficulty.

## 2 DREAMBENCH++

### 2.1 PROMPTING GPT FOR AUTOMATED & HUMAN-ALIGNED BENCHMARKING

It is challenging to obtain a solid quantitative understanding of generated models, especially when evaluating visual contents that rely on human evaluations (Lee et al., 2023; Ku et al., 2024). Thus, it is critical to achieve automated evaluation by utilizing multimodal GPT models, which are trained particularly in the principle of aligning with human preference (Ouyang et al., 2022; Christiano et al., 2017; OpenAI, 2024; 2023). This is evidenced by the recent progress achieved by Wu et al., which demonstrates that GPT-4V (OpenAI, 2023) can serve as a human-aligned text-to-3D generation evaluator. However, as pointed out by Zhang et al. and Ku et al., multimodal GPT models often fall short in evaluating personalized image generation—often more challenging when distinguishing *subtle difference* for concept preservation assessment using GPT—still underexplored. To tackle this issue, we detail how we systematically design the prompt of multimodal GPT (GPT-4o (OpenAI, 2024), by default) for human alignment reinforcement but also improve the reasoning progress that helps the GPT models to be more self-aligned, introduced as follows.

**Compare or rate?** There are typically two schemes for quantitatively evaluating generative models in human evaluations: *rating* and *comparison* (Zhang et al., 2023c; Zheng et al., 2023). The rating scheme requires human reviewers to assign an absolute score to each instance, while the comparison scheme asks human reviewers to express a relative preference among different instances. Though effective as the comparison scheme is when humans are involved, we find that there are two critical issues. **i)** *Positional Bias*: the scoring results of GPT-4V/GPT-4o is sensitive to the order in which images are presented (OpenAI, 2023; Wang et al., 2023a;b; Zhang et al., 2023c; Wu et al., 2024b; Zheng et al., 2023), making it unsuitable for comparison scheme. **ii)** *Quadratic Complexity*: As the number of methods increases, the number of essential evaluation runs for numerical rating increases linearly, while the number of comparative assessments increases quadratically. Therefore, direct numerical rating is more efficient and scalable when evaluating multiple methods. Hence, in this work, we adhere to the rating scheme, and we establish a *5-level rating scheme* where scores are integers ranging from 0 (very poor) to 4 (excellent).

**Evaluation Instructions** The evaluation instructions serve as the meta-prompting that describes overall tasks, which is shown in Fig. 3. As stated in Section 1, there are two fundamental quality criteria to be evaluated: i) *concept preservation* and ii) *prompt following*. For each aspect, we use a similar prompt template that contains ❶ **task description**, ❷ **scoring criteria explanation**, ❸ **scoring range definition**, and ❹ **format specification**. Only the scoring criteria are tailored for different tasks: for concept preservation evaluation, we prompt GPT to focus on *shape*, *color*, *texture*, and *facial features* (if applicable), while for prompt following evaluation we requested for focus on *relevance*, *accuracy*, *completeness* and *context*.

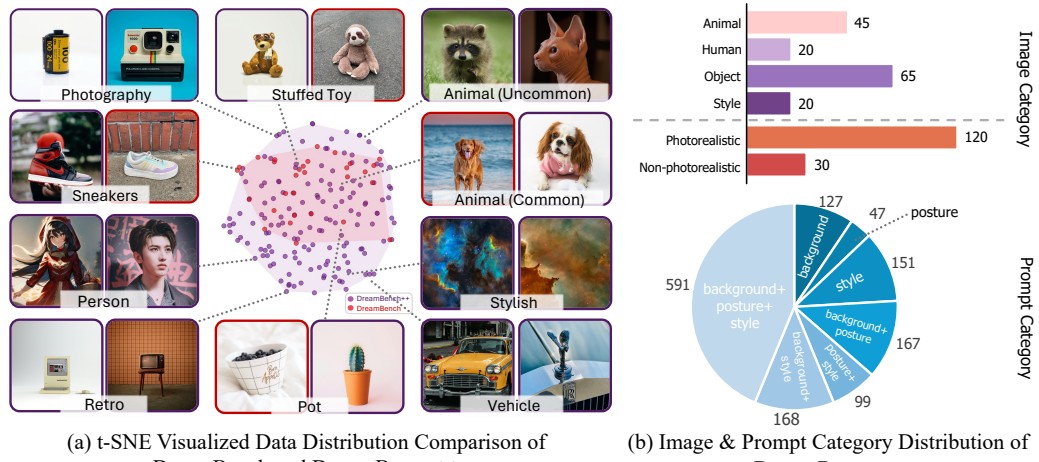

(a) t-SNE Visualized Data Distribution Comparison of DreamBench and DREAMBENCH++

(b) Image & Prompt Category Distribution of DREAMBENCH++

Figure 5: **Data distribution visualization.** (a) Images comparison between DreamBench and DREAMBENCH++ using t-SNE. (b) Image and prompt distribution of DREAMBENCH++.

**Reasoning Instructions** Given the evaluation instructions, it is crucial to reinforce the alignment with both the human instruction and itself to largely leverage the pretrained knowledge. To this end, we adopt a 2-step evaluation policy: **i)** *Internal Thinking*: Inspired by Self-Align (Sun et al., 2023), we introduce internal thinking to strengthen task understanding and instruction following capabilities. Specifically, we prompt the GPT model by asking if it understands the task or not and let it summarize the task. **ii)** *Summary & Planning*: According to the given internal thinking instruction, the GPT will summarize and plan for the evaluation task itself. It can also be viewed as a generalized form of chain-of-thought reasoning (Wei et al., 2022; Zhang et al., 2023d). See Fig. 3.

## 2.2 SCALING UP PERSONALIZED IMAGE GENERATION BENCHMARKING

Pioneering works like DreamBooth (Ruiz et al., 2023), SuTI (Chen et al., 2023c) and CustomConcept101 (Kumari et al., 2023) have successfully set up baseline datasets evaluating personalized image generation, and DREAMBENCH++ follows them to categorize images into three types: ❶ **objects**, ❷ **living subjects**, and ❸ **styles**. However, DreamBench's small scale and CustomConcept101's limited diversity may lead to biased evaluations, as some methods may converge well on its samples while performing unsatisfactorily on other data. To mitigate this, we scale up data by increasing both the number and diversity of images.

**Data Construction from Internet** The internet offers vast images, with many datasets built from it (Schuhmann et al., 2021; Jia et al., 2021). DREAMBENCH++ primarily sources images from Unsplash (uns), Rawpixel (raw), and Google Image Search (goo), along with authorized individual contributions. *Each image's copyright status has been verified for academic suitability.* As shown in Fig. 4, we collect and curate high-quality data to ensure robustness and diversity.

- **Keywords Generation** First, we generate 200 relevant keywords using GPT-4o and join them with the 200 most frequent keywords from Unsplash. After filtering out duplicated keywords, seven human annotators will extend the list to around 300 based on their interests.

- **Internet Images Collection** Given selected keywords, we retrieved images from Unsplash, Rawpixel, and Google Image Search. To filter out unsuitable images, SAM (Kirillov et al., 2023) identifies subject regions, discarding those with small subject areas. Human annotators remove images with noisy backgrounds. Curated images are then cropped to centralize the subject, yielding two images per keyword. Keywords without suitable images are discarded.

- **Prompt Generation** After image collection, 9 text prompts per image were generated using GPT-4o, designed to cover a range of difficulties: 4 prompts for ❶ **photorealistic** styles, 3 for ❷ **non-photorealistic** styles, and 2 for ❸ **complicated & imaginative** contents. To align with established evaluation methods, we use few-shot prompts selected from PartiPrompts (Yu et al., 2022). Human calibration ensures that all generated prompts are ethical and without flaws. As a result, the construction process finally yielded 150 high-quality images and 1,350 prompts.

Table 1: **Evaluation of personalized image generation models on DREAMBENCH++.** All scores are normalized to 0-1, and -I & -T represent image & text, respectively.

| Method | T2I Model | Concept Preservation | | | | Prompt Following | | |
|---|---|---|---|---|---|---|---|---|
| | | Human | GPT | DINO-I | CLIP-I | Human | GPT | CLIP-T |
| ● Textual Inversion | SD v1.5 | 0.316 | 0.378±0.0012 | 0.437 | 0.726 | 0.604 | 0.624±0.0033 | 0.302 |
| ● DreamBooth | SD v1.5 | 0.453 | 0.493±0.0012 | 0.544 | 0.753 | 0.679 | 0.721±0.0016 | 0.323 |
| ● DreamBooth LoRA | SDXL v1.0 | 0.571 | 0.597±0.0007 | 0.628 | 0.784 | 0.821 | 0.865±0.0007 | 0.341 |
| ● BLIP-Diffusion | SD v1.5 | 0.513 | 0.547±0.0010 | 0.649 | 0.823 | 0.577 | 0.495±0.0005 | 0.286 |
| ● Emu2 | SDXL v1.0 | 0.410 | 0.528±0.0016 | 0.539 | 0.763 | 0.641 | 0.689±0.0010 | 0.310 |
| ● IP-Adapter-Plus ViT-H | SDXL v1.0 | 0.755 | 0.833±0.0008 | 0.834 | 0.917 | 0.541 | 0.413±0.0005 | 0.282 |
| ● IP-Adapter ViT-G | SDXL v1.0 | 0.570 | 0.593±0.0018 | 0.667 | 0.855 | 0.688 | 0.640±0.0017 | 0.309 |

**Diversity Visualization** Internet images are numerous. However, there is a bias towards *photorealistic* styles. To diversify, various *non-photorealistic* styles are enlisted, and human annotators are tasked to gather images for each style, including *anime*, *sketches*, *traditional Chinese paintings*, *artworks*, and *cartoon characters from games*. Then, a manual selection process ensures a balanced distribution across subject classes and between photorealistic and non-photorealistic styles. In Fig. 5(a), we visualize the t-SNE (Van der Maaten & Hinton, 2008; Poličar et al., 2019) of images from DreamBench and DREAMBENCH++, which demonstrates the superiority of DREAMBENCH++ in diversity. Besides, Fig. 5(b) presents the detailed image distribution in DREAMBENCH++.

## 3 EXPERIMENTS

### 3.1 EXPERIMENTAL SETUP

**Reimplementation Details** We conduct experiments on two mainstream methods: i) ● *Fine-tuning-based methods*, including ❶ **Textual Inversion (TI) (Gal et al., 2023a)**, ❷ **Dream-Booth (Ruiz et al., 2023)**, and ❸ **DreamBooth LoRA (DreamBooth-L) (Ruiz et al., 2023; Hu et al., 2022)**; ii) ● *Encoder-based methods* that trains feature adaptation, including ❹ **BLIP-Diffusion (BLIP-D) (Li et al., 2023a)**, ❺ **Emu2 (Sun et al., 2024a)**, ❻ **IP-Adapter-Plus ViT-H (IP-Adapt.-P) (Ye et al., 2023)**, and ❼ **IP-Adapter ViT-G (IP-Adapt.) (Ye et al., 2023)**. All methods are based on base T2I models, including SD v1.5 (Rombach et al., 2022) and SDXL v1.0 (Podell et al., 2024). We stay true to the official implementations wherever possible and dedicate significant effort to parameter tuning for performance assurance on DreamBench, see details in Appendix B.

**Human Annotators** We employ 7 human annotators to score each instance in DREAMBENCH++ to obtain ground truth human preference data. We provide human annotators with sufficient training to ensure they fully understand the personalized T2I generation task and can provide *unbiased* and *discriminating* scores. *The scoring task and scheme given to humans are identical to those used for GPT, as described in Section 2.* The GPT results and human results are isolated to avoid hindsight bias. Additionally, we ensure that each instance is rated by *at least two humans* to reduce noise.

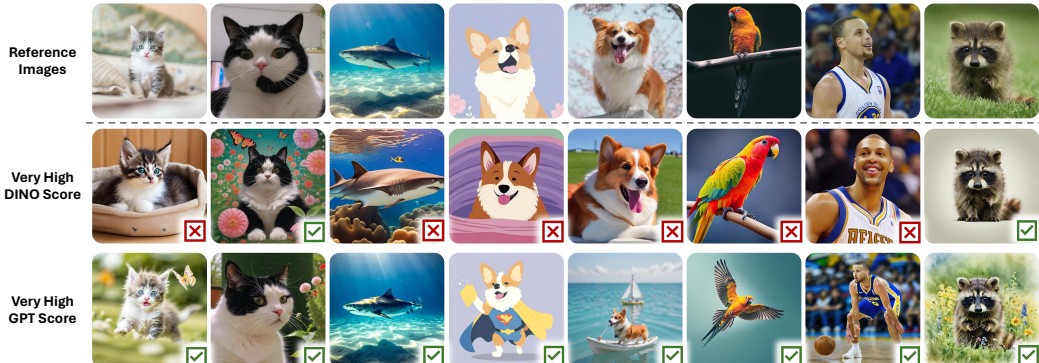

Figure 6: **Comparison between images of high DINO score and high GPT-4o score.** Instances with high human scores are ticked, and those with low human scores are crossed. DINO tends to yield high scores to images that preserve overall shape but do not put much weight on color, texture, and facial features, leading to frequent contradiction with human preference.

Table 2: **DREAMBENCH++ leaderboard.** Both scores for concept preservation (CP) and prompt following (PF) are presented and divided by established concept and prompt categories. The models are ranked by the product of CP and PF scores (CP·PF).

| Method | T2I Model | Concept Preservation | | | | | Prompt Following | | | | CP·PF | CP/PF |
|---|---|---|---|---|---|---|---|---|---|---|---|---|
| | | Animal | Human | Object | Style | Overall | Realistic | Style | Imaginative | Overall | | |
| • DreamBooth LoRA | SDXL v1.0 | 0.751 | 0.311 | 0.543 | 0.718 | 0.598 | 0.898 | 0.895 | 0.754 | 0.865 | 0.517 | 0.69 |
| • IP-Adapter ViT-G | SDXL v1.0 | 0.667 | 0.558 | 0.504 | 0.752 | 0.593 | 0.743 | 0.632 | 0.446 | 0.640 | 0.380 | 0.93 |
| • Emu2 | SDXL v1.0 | 0.670 | 0.546 | 0.447 | 0.454 | 0.528 | 0.732 | 0.719 | 0.560 | 0.690 | 0.364 | 0.77 |
| • DreamBooth | SD v1.5 | 0.640 | 0.199 | 0.488 | 0.476 | 0.494 | 0.789 | 0.775 | 0.504 | 0.721 | 0.356 | 0.69 |
| • IP-Adapter-Plus ViT-H | SDXL v1.0 | 0.900 | 0.845 | 0.759 | 0.912 | 0.833 | 0.502 | 0.384 | 0.279 | 0.413 | 0.344 | 2.02 |
| • BLIP-Diffusion | SD v1.5 | 0.673 | 0.557 | 0.469 | 0.507 | 0.547 | 0.581 | 0.510 | 0.303 | 0.495 | 0.271 | 1.11 |
| • Textual Inversion | SD v1.5 | 0.502 | 0.358 | 0.305 | 0.358 | 0.378 | 0.671 | 0.686 | 0.437 | 0.624 | 0.236 | 0.61 |

## 3.2 MAIN RESULTS

**Quantitative & Qualitative Analysis** Table 1 shows the overall evaluation results, including human and GPT-4o rating scores. The results show that: **i)** DREAMBENCH++ *aligns better with humans than DINO or CLIP models*. Driven by our dedicatedly-designed prompts, GPT-4o used by DREAMBENCH++ yields impressive alignment with humans. This is because humans and DREAMBENCH++ are all advanced in evaluating facial and textural characters and producing scores with a balanced consideration. **ii)** DINO-I and CLIP-I yield significant divergence from humans in evaluating concept preservation. This could be because DINO/CLIP scores show a preference for images that preserve shapes or overall styles (see Fig. 6). **iii)** Traditional CLIP-T scores are as effective as DREAMBENCH++ in evaluating prompt following, showing strong alignment with humans. See qualitative results in Appendix A for an intuitive understanding of evaluated models.

**Leaderboard** Table 2 shows the leaderboard results with respect to the concept and prompt categories defined in Section 2. Note that: **i)** the human category shows the lowest average score of 0.482, which is -0.204 lower than the highest average score of animal. This category is very challenging in terms of concept preservation because due to facial details, and many works are conducted specifically on it (Wang et al., 2024b; Xiao et al., 2023; Valevski et al., 2023; Yan et al., 2023). **ii)** The object is also a relatively difficult category due to object diversity. In contrast, animals within the same category often share a strong visual similarity. **iii)** There exists a negative correlation between concept preservation and prompt following. The primary aim of personalized T2I evolution is to identify the Pareto optimum that balances both. **iv)** The CP/PF ratio reflects the over-fitting issue. A higher ratio indicates excessive adherence to the reference at the cost of prompt alignment, while a lower ratio suggests better prompt adherence but weaker concept preservation.

**Abaltion Study** Table 3 shows the ablation study of the prompt design influences on alignment measured by mean Krippendorff's alpha value (Hayes & Krippendorff, 2007). We observe that: **i)** The proposed prompt designs are all necessarily effective, demonstrating the superiority of the prompting method in DREAMBENCH++. For example, removing the proposed internal thinking leads to a significant drop, indicating the effectiveness of self-alignment. **ii)** The capability of the multimodal GPT used is scalable. This shows that DREAMBENCH++ has the potential to be improved in the future. **iii)** Some human prior knowledge, such as reminding the GPT not to consider background when assessing visual concept preservation, leads to performance degradation.

Table 3: **Ablation study of prompt design.** H, G, D, and C represent Human, GPT-4o, DINO Score, and CLIP Score, respectively. H-H value is also calculated to illustrate human self-alignment.

| Method
T2I Model | TI
SD v1.5 | DreamBooth
SD v1.5 | DreamBooth-L
SDXL v1.0 | BLIP-D
SD v1.5 | Emu2
SDXL v1.0 | IP-Adapt.-P
SDXL v1.0 | IP-Adapt.
SDXL v1.0 |
|---|---|---|---|---|---|---|---|
| *Concept Preservation* $\mathrm{Kd}_{\bar{O}}$ | | | | | | | |
| **H-H** | 0.685 | 0.647 | 0.656 | 0.613 | 0.746 | 0.602 | 0.591 |
| **G-H** | 0.544±0.014 | 0.596±0.003 | 0.641±0.007 | 0.362±0.017 | 0.669±0.005 | 0.366±0.017 | 0.458±0.002 |
| - Internal Thinking | -0.040 | -0.023 | -0.012 | +0.001 | -0.045 | -0.038 | -0.008 |
| - Scoring Criteria | -0.125 | -0.116 | -0.093 | -0.158 | -0.103 | -0.227 | -0.166 |
| - Scoring Range | -0.038 | -0.017 | -0.027 | -0.006 | -0.016 | -0.009 | -0.017 |
| + Human Prior | -0.033 | -0.022 | -0.006 | -0.015 | -0.022 | +0.009 | -0.019 |
| + GPT4V | -0.105 | -0.067 | -0.131 | -0.180 | -0.016 | -0.301 | -0.250 |
| *Prompt Following* $\mathrm{Kd}_{\bar{O}}$ | | | | | | | |
| **H-H** | 0.475 | 0.516 | 0.469 | 0.619 | 0.441 | 0.576 | 0.509 |
| **G-H** | 0.461±0.007 | 0.506±0.002 | 0.402±0.001 | 0.541±0.003 | 0.422±0.011 | 0.484±0.006 | 0.531±0.006 |
| - Internal Thinking | -0.013 | +0.004 | -0.032 | -0.002 | -0.014 | +0.012 | -0.002 |
| - Scoring Criteria | -0.025 | -0.012 | -0.009 | -0.012 | -0.018 | -0.017 | -0.013 |
| - Scoring Range | -0.010 | -0.013 | -0.011 | +0.043 | -0.038 | +0.060 | +0.036 |
| + GPT4V | -0.010 | +0.012 | 0.000 | -0.111 | -0.007 | -0.161 | -0.134 |

# 4 DISCUSSIONS

## 4.1 IS DREAMBENCH++ ALIGNED WITH HUMANS?

Table 4 shows a more rigorous study of human alignment level using the mean Pearson correlation value (Pearson, 1895). The results show that **DREAMBENCH++ is a highly human-aligned benchmark**. Notably, DREAMBENCH++ achieves **83.31%** and **98.71%** evaluation consistency with human's evaluation in concept preservation and prompt following capabilities, respectively. This result is **+32.59%** and **+37.23%** higher than traditional DINO and CLIP metrics.

Table 4: **The human alignment degree among different evaluation metrics**, measured by Pearson correlation value. H, G, D, and C represent Human, GPT-4o, DINO Score, and CLIP Score, respectively. H-H value is also calculated to illustrate human self-alignment.

| Method | T2I Model | Concept Preservation $PC_{\bar{O}}$ | | | | Prompt Following $PC_{\bar{O}}$ | | |
|---|---|---|---|---|---|---|---|---|
| | | H-H | G-H | D-H | C-H | H-H | G-H | C-H |
| • Textual Inversion | SD v1.5 | 0.685 | 0.576±0.005 | 0.499±0.005 | 0.546±0.002 | 0.506 | 0.491±0.004 | 0.367±0.004 |
| • DreamBooth | SD v1.5 | 0.647 | 0.611±0.002 | 0.547±0.003 | 0.531±0.012 | 0.531 | 0.526±0.028 | 0.302±0.012 |
| • DreamBooth LoRA | SDXL v1.0 | 0.657 | 0.640±0.015 | 0.474±0.030 | 0.513±0.015 | 0.474 | 0.417±0.029 | 0.211±0.018 |
| • BLIP-Diffusion | SD v1.5 | 0.614 | 0.386±0.001 | 0.158±0.005 | 0.274±0.000 | 0.637 | 0.602±0.015 | 0.420±0.004 |
| • Emu2 | SDXL v1.0 | 0.745 | 0.733±0.002 | 0.721±0.011 | 0.701±0.013 | 0.445 | 0.436±0.009 | 0.308±0.012 |
| • IP-Adapter-Plus ViT-H | SDXL v1.0 | 0.603 | 0.407±0.022 | -0.043±0.010 | 0.132±0.001 | 0.579 | 0.581±0.010 | 0.334±0.011 |
| • IP-Adapter ViT-G | SDXL v1.0 | 0.591 | 0.464±0.007 | 0.060±0.024 | 0.165±0.027 | 0.511 | 0.583±0.015 | 0.325±0.034 |
| **Ratio$_{\bar{O}}$** | | 100% | 83.31% | 50.72% | 60.98% | 100% | 98.17% | 61.48% |

## 4.2 IS DATA DIVERSITY NECESSARY?

To assess the importance of diverse data, we compare results on DreamBench and DREAMBENCH++ using DINO and CLIP metrics. Table 5 shows that **the diverse data in DREAMBENCH++ is key to unbiased evaluation.** While overall results are consistent, TI, DreamBooth, and Emu2 show notable score drops. These methods perform well on natural images and simple text but struggle with complex or stylized prompts and anime references, see Fig. 7.

Table 5: **DreamBench and DREAMBENCH++ results comparison with traditional metrics.** *Unlike DreamBench, DREAMBENCH++ uses a single reference image per instance; thus, the training steps and learning rate of • fine-tuning-based methods are slightly reduced to avoid overfitting.

| Method | T2I Model | DreamBench | | | DREAMBENCH++ | | |
|---|---|---|---|---|---|---|---|
| | | DINO-I | CLIP-I | CLIP-T | DINO-I | CLIP-I | CLIP-T |
| • Textual Inversion* | SD v1.5 | 0.557 | 0.753 | 0.259 | 0.437 | 0.726 | 0.302 |
| • DreamBooth* | SD v1.5 | 0.678 | 0.786 | 0.301 | 0.544 | 0.753 | 0.323 |
| • DreamBooth LoRA* | SDXL v1.0 | 0.646 | 0.769 | 0.325 | 0.628 | 0.784 | 0.341 |
| • BLIP-Diffusion | SD v1.5 | 0.630 | 0.784 | 0.293 | 0.649 | 0.823 | 0.286 |
| • Emu2 | SDXL v1.0 | 0.753 | 0.842 | 0.283 | 0.539 | 0.763 | 0.310 |
| • IP-Adapter-Plus ViT-H | SDXL v1.0 | 0.846 | 0.902 | 0.272 | 0.834 | 0.917 | 0.282 |
| • IP-Adapter ViT-G | SDXL v1.0 | 0.681 | 0.835 | 0.295 | 0.667 | 0.855 | 0.309 |

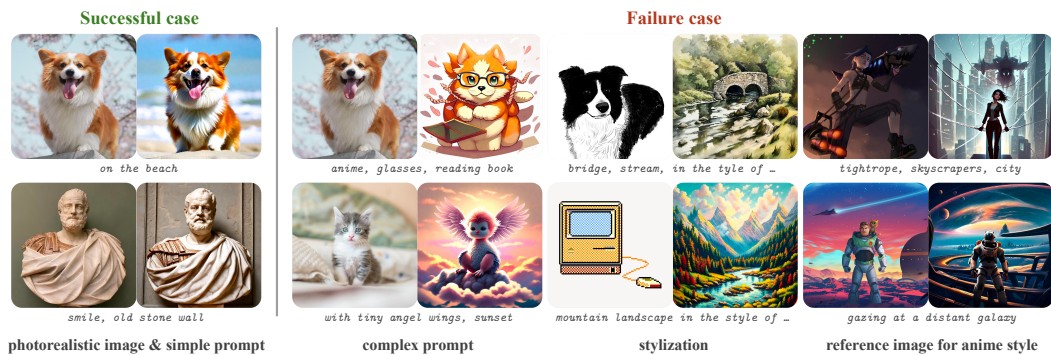

**Successful case**      **Failure case**

*on the beach*    *anime, glasses, reading book*    *bridge, stream, in the tyle of ...*    *tightrope, skyscrapers, city*

*smile, old stone wall*    *with tiny angel wings, sunset*    *mountain landscape in the style of ...*    *gazing at a distant galaxy*

**photorealistic image & simple prompt**    **complex prompt**    **stylization**    **reference image for anime style**

Figure 7: **Case study of successful and failure case on DREAMBENCH++.** The left images are reference images and the right images are results generated by Emu2, DreamBooth, and TI.

### 4.3 CAN WE USE FREE LUNCH TO IMPROVE DREAMBENCH++ EVALUATION?

Table 6 shows the result of utilizing free lunch techniques, including chain-of-thought (CoT) (Wei et al., 2022) with GPT-4 articulating reasoning process and In-Context Learning (ICL) (Alayrac et al., 2022; Brown et al., 2020) with human-written few-shot examples.

**Chain-of-Thought:** i) CoT is effective in evaluating prompt following capability. Through CoT, the model more accurately discerns the significance of phrases such as "morphs into a mythical dragon", allowing it to assign a more appropriate evaluation score. ii) CoT does not bring improvement in concept preservation evaluation. We argue that CoT may shift attention to unnecessarily important background or texture information, as shown in Fig. 8.

Table 6: **Study of Chain-of-Though (CoT) and In-context Learning (ICL) on human alignment.**

| Method
T2I Model | TI
SD v1.5 | DreamBooth
SD v1.5 | DreamBooth-L
SDXL v1.0 | BLIP-D
SD v1.5 | Emu2
SDXL v1.0 | IP-Adapt.-P
SDXL v1.0 | IP-Adapt.
SDXL v1.0 |
|---|---|---|---|---|---|---|---|
| **H-H** | 0.685 | 0.647 | 0.656 | 0.613 | 0.746 | 0.602 | 0.591 |
| **w/o CoT & w/o ICL** | 0.544 | 0.596 | 0.641 | 0.362 | 0.669 | 0.366 | 0.458 |
| **+ 1 shot ICL** | -0.046 | -0.019 | -0.043 | +0.013 | -0.028 | -0.098 | -0.066 |
| **+ 2 shot ICL** | -0.042 | -0.023 | -0.022 | -0.033 | -0.036 | -0.085 | -0.054 |
| **w/ CoT & w/o ICL** | 0.510 | 0.576 | 0.602 | 0.329 | 0.644 | 0.359 | 0.418 |
| **+ 1 shot ICL** | -0.040 | -0.008 | -0.009 | -0.020 | -0.035 | -0.145 | -0.086 |
| **+ 2 shot ICL** | -0.030 | -0.002 | -0.002 | -0.051 | -0.031 | -0.155 | -0.082 |

**In-Context Learning:** ICL counterintuitively leads to a drop in alignment. This could be attributed to the patching scheme, sample selection, or inherent bias within GPT-4o, making it non-trivial to prompt effectively. Thus, we provide our detailed prompt and hope to inspire future works.

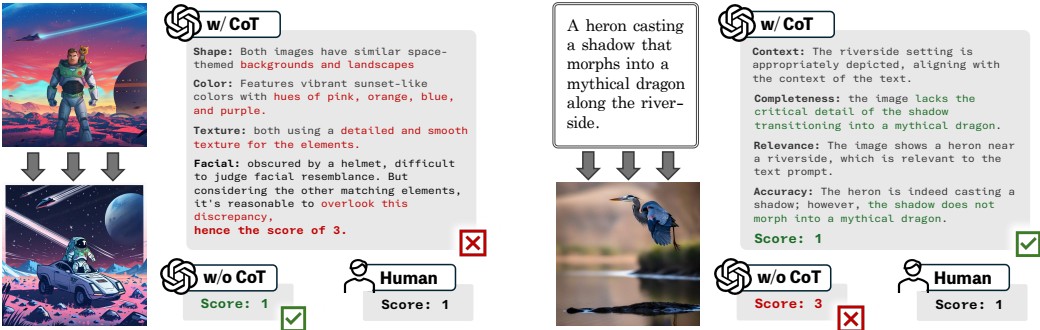

Figure 8: **Case study on CoT Prompting**. We find that (a) CoT prompting can improve text following evaluation by recognizing important parts of the prompt. (b) However, it may also hinder visual concept preservation by drifting GPT's attention away from the subject.

### 4.4 HOW TRANSFERABLE IS DREAMBENCH++ ACROSS MLLMS?

Table 7 reveals: i) DREAMBENCH++ demonstrates robust generalization across different multi-modal LLMs. The consistency ratios remain stable across various models, with Claude-3.5 achieving 74.00% and GPT-4o variants ranging from 67.22% to 84.23%. ii) The performance of MLLMs is continuously improving, as shown by the increasing Human-MLLM consistency in newer versions of GPT-4o (from 79.47% to 84.23%). This suggests that the accuracy of DREAMBENCH++ evaluations may further improve as models advance.

Table 7: **Comparison of Human-MLLM consistency ratios**. Ratio: the overall alignment between model evaluations and human evaluations, calculated as the average proportion of H-H consistency.

| Method
T2I Model | TI
SD v1.5 | DreamBooth
SD v1.5 | DreamBooth-L
SDXL v1.0 | BLIP-D
SD v1.5 | Emu2
SDXL v1.0 | IP-Adapt.-P
SDXL v1.0 | IP-Adapt.
SDXL v1.0 | Ratio |
|---|---|---|---|---|---|---|---|---|
| **H-H (Ground Truth)** | 0.685 | 0.647 | 0.656 | 0.613 | 0.746 | 0.602 | 0.591 | 100% |
| **GPT-4o-2024-05-13 (DREAMBENCH++)** | 0.544 | 0.596 | 0.641 | 0.362 | 0.669 | 0.366 | 0.458 | 79.47% |
| **GPT-4o-2024-08-06** | 0.558 | 0.625 | 0.645 | 0.383 | 0.708 | 0.427 | 0.502 | 84.23% |
| **GPT-4o-mini-2024-07-18** | 0.496 | 0.575 | 0.538 | 0.274 | 0.702 | 0.238 | 0.289 | 67.22% |
| **Claude-3.5-Sonnet-20241022** | 0.500 | 0.583 | 0.623 | 0.354 | 0.625 | 0.278 | 0.427 | 74.00% |
| **Gemini-1.5-Pro-001** | 0.487 | 0.547 | 0.514 | 0.267 | 0.653 | 0.253 | 0.202 | 63.04% |

## 5 RELATED WORKS

**Personalized Image Generation**   aims to preserve concept consistency while accommodating the diverse contexts suggested by the instructions. In general, it can be traced back to early efforts on pixel-to-pixel (Pix2Pix) translation where the personalization orientation is free-form texts (Brooks et al., 2023; Tumanyan et al., 2023; Parmar et al., 2023) or predefined translation across styles, seasons, species, or plants, *etc* (Isola et al., 2017; Zhu et al., 2017; Zhang et al., 2023a; Wang et al., 2018; Saharia et al., 2022a). Modern efforts go beyond Pix2Pix translation toward a free-form image generation conditioned on both reference images and prompts. Some works focus on fine-tuning techniques that turn a general T2I model into a specialist personalization model (Gal et al., 2023a; Ruiz et al., 2023; Kumari et al., 2023; Sohn et al., 2023; Park et al., 2024) using LoRA (Hu et al., 2022) or contrastive learning (Zhang et al., 2022; He et al., 2020), learning the subject or style information by reconstructive autoencoding (Vincent et al., 2008; Dong et al., 2023). However, the necessity to fine-tune for new subjects limits their scalability. In contrast, encoder-based methods can generate subject-guided or style-guided images or edit images following prompts with one shot. Encoder- or adapter-based methods (Zheng et al., 2024; Ye et al., 2023; Wei et al., 2023; Li et al., 2023a; Jia et al., 2023; Gal et al., 2023b; Chen et al., 2023b; Wang et al., 2024a;b) train an encoder to encode the conditional image into embeddings, which are integrated into cross-attention mechanism in the diffusion process (Ho et al., 2020; Song et al., 2021a; Nichol & Dhariwal, 2021; Song et al., 2021b). Adapter-free methods (Lv et al., 2024; Liu et al., 2023d; Hertz et al., 2023b;a; Brooks et al., 2023) extract the information, such as attention maps (Hertz et al., 2023a) from reference images, and fuse them into the image generation process. Furthermore, multimodal large language models (MLLMs) that are trained on extensive multimodal sequences can also serve as general foundation models (Dong et al., 2024; Sun et al., 2024a; Pan et al., 2024; Ge et al., 2024). Additionally, some works (Wang et al., 2024b; Valevski et al., 2023; Yan et al., 2023; Ye et al., 2023; Xiao et al., 2023) focus on facial feature preservation.

**Benchmarking Image Generation**   involves a variety of metrics that focus on different aspects. Inception Score (Salimans et al., 2016) and FID (Heusel et al., 2017) judge image quality, while LPIPS (Zhang et al., 2018), DreamSim (Fu et al., 2023), CLIP-I (Radford et al., 2021), and DINO Score (Caron et al., 2021) measure perceptual similarity. In text-guided generation, prompt-image alignment can be assessed by CLIP-T (Radford et al., 2021), CLIPScore (Hessel et al., 2021), and BLIP Score (Li et al., 2022; 2023b). However, these metrics often fall short of reflecting human perception. To address this, human-aligned metrics (Ku et al., 2024; Xu et al., 2023; Lee et al., 2023) have been introduced, offering a more perceptive evaluation. Yet, they face limitations in scaling with the pace of new model developments. Thus, the necessity for automated and sustainable evaluation methods has emerged, with some (Xu et al., 2023; Liang et al., 2023b; Guo et al., 2024) leveraging reward-model-based methods to encode human preferences, while others (Ku et al., 2023; Cho et al., 2023; Wu et al., 2024b; Zhang et al., 2023c; Hu et al., 2023; Lu et al., 2023) use multimodal  (Brown et al., 2020; Reid et al., 2024; Anil et al., 2023; Liu et al., 2023b) to automate the process and better mirror human tastes. While MLLM-based methods show promise in aligning with human preferences (Zhang et al., 2023c; Wu et al., 2024b; Huang et al., 2023; Cho et al., 2024), automated personalized evaluation remains an unresolved issue. VIEScore (Ku et al., 2023) assesses image generation quality by prompting GPT-4V (OpenAI, 2023) and LLaVA (Liu et al., 2023b;a), but is limited to four models in subject-driven tasks and obtains suboptimal results. Meanwhile, Dreambench (Ruiz et al., 2023), a common benchmark for personalized generative evaluation, only consists of 30 simple objects and lacks diversity comprehensiveness.

## 6 CONCLUSIONS

This paper introduces DREAMBENCH++, a human-aligned personalized image generation benchmark. Extensive and comprehensive experiments show significant advantages in dataset diversity and complexity, along with metrics that align with human preferences. In addition, we offer insights into prompt design for advanced multimodal GPTs, emphasizing the potential and challenges of enhancing GPT evaluation through chain-of-thought prompting and in-context learning. Our work aims to support future research on personalized image generation by providing a human-aligned benchmark and heuristics in utilizing advanced multimodal GPTs in visual evaluation.

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

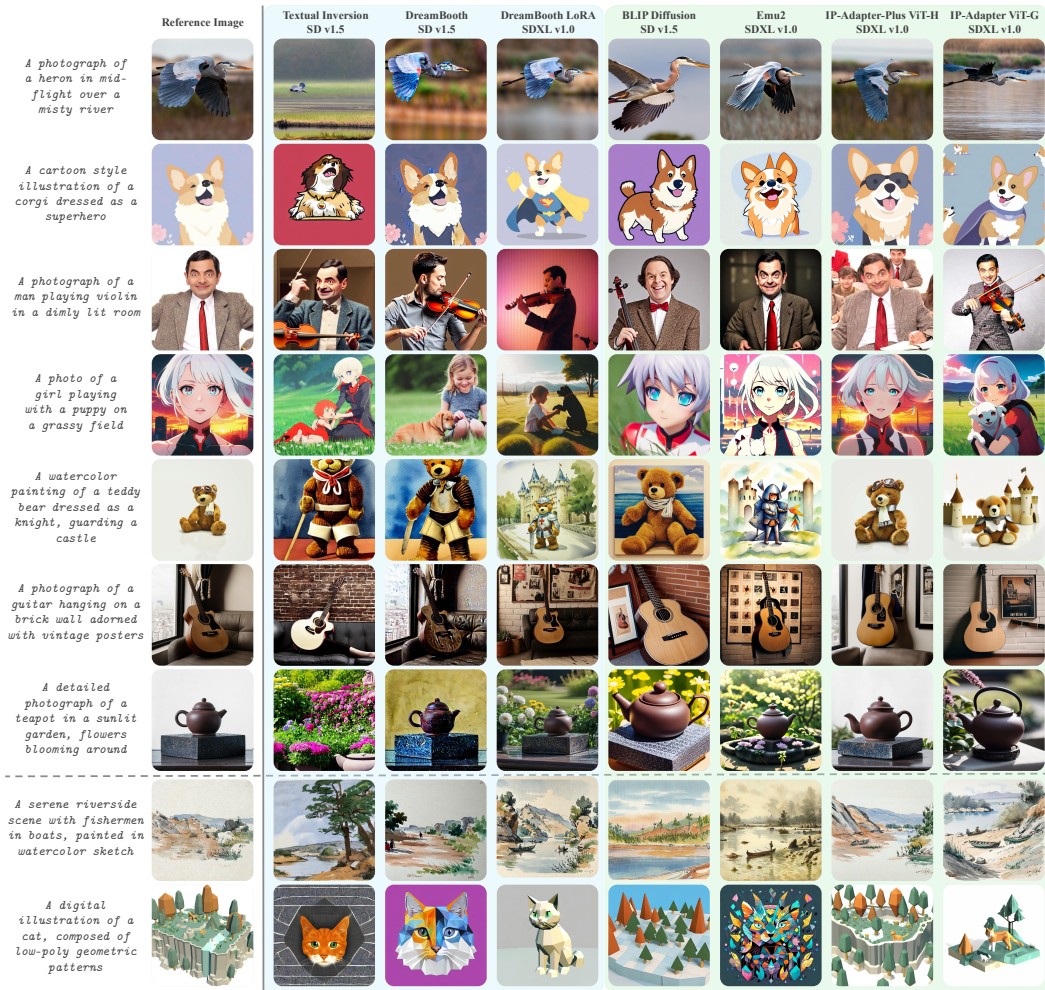

Figure 9: **A qualitative study of different methods on DREAMBENCH++**. We demonstrate the generation quality of different methods on our DREAMBENCH++, including animals, humans, objects, and style, with photo and non-photo-realistic examples. The blue block highlights fine-tuning-based methods, and the green block highlights encoder-based methods. Instances above the dotted line are evaluated for subject preserving, and instances below are evaluated for style preserving.

## A  QUALITATIVE ANALYSIS

With a more comprehensive and diverse collection of images, we have discovered numerous intriguing characteristics of these generation methods, as illustrated in Fig. 9, that were not apparent on existing datasets such as DreamBench. Specifically, we observe that: **i)** Fine-tuning-based methods outperform encoder-based methods on images containing more *subject-oriented* information, such as an animal or object, as they preserve more intricate details in the generated images. However, for images containing a person, fine-tuning-based methods often fail to preserve facial and clothing features. This suggests that the personalized generation of human images is more demanding for visual concept preservation than for textual following capabilities, which is an advantage for encoder-based methods. **ii)** However, for *style-oriented* cases when subject details are less critical, encoder-based methods perform better than fine-tuning-based methods. This further highlights the strengths of encoder-based methods in that they are more adept at recognizing and extracting high-level visual semantics, including overall shape, style, and thematic features.

Table 8: **Training hyperparameters on DreamBench and DREAMBENCH++**. BS: batch size, LR: learning rate, Steps: training steps.

| | | DreamBench | | | DREAMBENCH++ | | |
|---|---|---|---|---|---|---|---|
| **Method** | **T2I Model** | **BS** | **LR** | **Steps** | **BS** | **LR** | **Steps** |
| Textual Inversion | SD v1.5 | 4 | 5e-4 | 3000 | 1 | 5e-4 | 3000 |
| Dreambooth | SD v1.5 | 1 | 2.5e-6 | 1000 | 1 | 2.5e-6 | 250 |
| Dreambooth LoRA | SDXL v1.0 | 4 | 5e-5 | 500 | 1 | 5e-5 | 500 |

## B  IMPLEMENTATION DETAILS

The configurations for the training hyperparameters used in training-based methods on DreamBench and DREAMBENCH++, are detailed in Table 8. During the inference stage, all methods employ a `guidance_scale` of 7.5 and execute 100 inference steps, with the exception of Emu2, which uses a `guidance_scale` of 3 and performs 50 inference steps. Furthermore, BLIP-Diffusion and IP-Adapter incorporate negative prompts, as demonstrated in Table 9. Specifically, IP-Adapter includes an additional parameter, `ip_adapter_scale`, set at 0.6.

Table 9: **Negative Prompt Templates**

| Method | T2I Model | Negative Prompt |
|---|---|---|
| BLIP-Diffusion | SD v1.5 | over-exposure, under-exposure, saturated, duplicate, out of frame, lowres, cropped, worst quality, low quality, jpeg artifacts, morbid, mutilated, ugly, bad anatomy, bad proportions, deformed, blurry, duplicate |
| IP-Adapter ViT-G | SDXL v1.0 | deformed, ugly, wrong proportion, low res, bad anatomy, worst quality, low quality |
| IP-Adapter-Plus ViT-H | SDXL v1.0 | deformed, ugly, wrong proportion, low res, bad anatomy, worst quality, low quality |

We dedicate significant effort to tuning hyper-parameters to ensure that the performance of each method on DreamBench is consistent with results reported in original papers. Table 10 shows the results of our reproduction are comparable to or even better than the official results.

Table 10: **Reproduced results**. Our reproduction is comparable to or better than the official results. N/A denotes that the official paper does not report the corresponding results.

| Method | T2I Model | DINO-I | | CLIP-I | | CLIP-T | |
|---|---|---|---|---|---|---|---|
| | | **Official** | **Reproduction** | **Official** | **Reproduction** | **Official** | **Reproduction** |
| • Textual Inversion | SD v1.5 | 0.569 | 0.557 | 0.780 | 0.753 | 0.255 | 0.259 |
| • DreamBooth | SD v1.5 | 0.688 | 0.678 | 0.803 | 0.786 | 0.305 | 0.301 |
| • DreamBooth LoRA | SDXL v1.0 | N/A | 0.646 | N/A | 0.769 | N/A | 0.325 |
| • BLIP-Diffusion | SD v1.5 | 0.594 | 0.630 | 0.779 | 0.784 | 0.300 | 0.293 |
| • Emu2 | SDXL v1.0 | 0.766 | 0.753 | 0.850 | 0.842 | 0.287 | 0.283 |
| • IP-Adapter-Plus ViT-H | SDXL v1.0 | N/A | 0.846 | N/A | 0.902 | N/A | 0.272 |
| • IP-Adapter ViT-G | SDXL v1.0 | N/A | 0.681 | N/A | 0.835 | N/A | 0.295 |

## C  ADDITIONAL DISCUSSIONS

### C.1  ARE MULTIPLE IMAGES FOR EACH INSTANCE NECESSARY?

In practice, multiple reference images are unnecessary for personalized image generation: **i)** The limited availability of reference images during daily usage makes single-image personalization more relevant. **ii)** Fine-tuning methods perform well with just one image, as shown in Appendix A.

### C.2  WHICH TECHNICAL ROUTE PERFORMS BETTER?

Our comprehensive analysis reveals distinct advantages of different technical routes across various generation scenarios. For concept preservation, encoder-based methods demonstrate superior performance in human-centric generation, particularly in preserving precise facial features where fine-tuning approaches often struggle. This aligns with the recent trend in Identity-Preserving Generation tasks. However, fine-tuning methods show stronger capabilities in handling common objects and standardized appearances, while encoder-based approaches provide more consistent performance across diverse object types.

The choice of base model significantly impacts generation quality. SDXL exhibits better concept preservation and prompt following capabilities compared to SD v1.5, primarily due to its more sophisticated text encoder architecture and diverse training data. Regarding prompt adherence, fine-tuning methods generally outperform encoder-based approaches by better preserving the original text-image conditional distribution through direct learning from image-text pairs. In contrast, encoder-based methods may face challenges in maintaining precise prompt alignment due to potential disruptions during feature injection.

These findings suggest that the optimal technical route depends heavily on the specific application scenario. For applications prioritizing human identity preservation, encoder-based methods are preferable. However, for use cases requiring precise prompt following or handling common objects, fine-tuning approaches might be more suitable. This understanding helps guide future research directions in personalized image generation.

## D  LIMITATION & FUTURE WORK

Human-aligned evaluation & benchmarking is an emerging but challenging direction, and we have only made preliminary attempts at personalized image generation. Moreover, our evaluation results heavily rely on the advancements of multimodal large language models and require carefully designed system prompts. We believe that as visual world models continue to develop, the evaluation performance will be further optimized. Our future work will focus on more applications with human-aligned evaluation, such as multimodal intelligence (Wei et al., 2024a; Zhu et al., 2025; Wei et al., 2024d;c; Zhu et al., 2024; Liu et al., 2024; Chen et al., 2024; Wei et al., 2024b; Gao et al., 2024a;b), audio generation (Gao et al., 2024c), 3D generation (Poole et al., 2023; Qi et al., 2023; Liu et al., 2023c; Gafni et al., 2022; Gao et al., 2023), video generation (Ho et al., 2022; Singer et al., 2023; Blattmann et al., 2023; Zhou et al., 2025), autonomous driving (Han et al., 2024; Li et al., 2023c; Zhang et al., 2023b), and even embodied visual intelligence (Goyal et al., 2022; Liang et al., 2023a; Brohan et al., 2023; Driess et al., 2023; Qi et al., 2025; Zhao et al., 2024; He et al., 2024b; Yu et al., 2024; 2025).

## BROADER IMPACT

Powerful as the T2I generative models pretrained on large-scale web-scraped data, the models may be misused as illegal or unethical tools for generating NSFW content. This potential impact can also be brought by personalized T2I models as they are typically built on the pretrained T2I foundation models. As a result, it is critical to use tools such as NSFW detectors to avoid such content during both usage and evaluation. For example, the data used for evaluation must avoid the NSFW content by data filtering. In this paper, such contents are filtered out by human annotators.

