# OpenReview forum: "DreamBench++: A Human-Aligned Benchmark for Personalized Image Generation"
_ICLR.cc/2025/Conference — ICLR 2025 Poster_

### Official Review · Reviewer_RNsM · 2024-10-29

**Soundness:** 3
**Presentation:** 3
**Contribution:** 2
**Rating:** 6
**Confidence:** 4

**Summary:**

This paper contributed a benchmark and a dataset for evaluating personalized image generation task, also an automated evaluating metric using GPT to address the issues of cost inefficient human evaluation, and also the lack of diversity in previous dataset/benchmark works. It included 7 models ( 3 more than prior works), and developed an semi-automated dataset creation pipeline. It also expanded dataset to 5x of images and 54x of prompts, and proposed the use of automated evaluating metric (GPT), with different settings studied (e.g. COT and ICL). The main goal of this work is to comprehensively evaluate these models to figure out which technical route is superior and where to head.

**Strengths:**

S1) Fruitful discussion in diversity study, showcasing that models are good at certain types of generation. (e.g. all models generally perform better in Animal and Style due to sensitivity to facial details and diverse object categories). This aligns with the goal to figure out which model is superior in certain types of generation.

S2) Well-organized study in showcasing the impact of prompt design including COT, ICL, Internal thinking etc.. when comes to automated evaluation with GPT.

**Weaknesses:**

W1) In Section 3.2 the authors claimed that "Table 1 results show that DreamBench++ aligns better with humans than DINO or CLIP models...", which is not very convincing. While it makes sense to show that the ranking order from GPT is more correlated to Human compared to DINO-I and CLIP-I, It is expected to show the correlation (Spearman / Pearson). I would suggest the authors to add a table regarding correlation between Human and GPT, and between Human and DINO/CLIP, supporting the claim. Also It seems a bit odd to use Krippendorff's Alpha to compare human ratings are other rating (GPT/DINO/CLIP) in Table 4. The two scales are inherently different and the meaning of ratings are different. Spearman / Pearson correlation would be a better metric for cross-scale reliability. Krippendorff's Alpha would be suitable for H-H as showing the inter-rater reliability.

W2) The paper would have been much stronger if the authors included a section to discuss which model performs the best in certain types of generation and how is it associated with certain technical route. This will be more aligned to the purposed goal "figure out which technical route is superior and where to head."

W3) Minor confusion when reading the table 1 and 3 directly. For example, Table 3 does not mention what exactly are the score values are. Please extend table captions to explain what are the values in the revised version.

W4) Please highlight the best model in each categories from Table 2 in the revised version. Maybe a leaderboard showcasing the best models in each category.

**Questions:**

Q1) Any design rationale on the score scale? Why not use a score in scale from 0 to 1? This would be more aligned with the range of traditional metrics (e.g. CLIPScore, LPIPS, DINO etc.)?

Q2) How do the authors verify that the automated evaluation with GPT results are making sense (such that the reasoning fully reflects the score)?

Q3) Please share the analysis of the costs and time needed for the human annotation, and how many data instance were annotated in total. Will the human annotated data be released?

Q4) Authors might want to consider adding more models for comparison, including DisenBooth(ICLR 2024), EZIGEN (ArXiv 2024),

---
I believe this study will interest a broad audience. However, the contribution feels somewhat limited, and the discussion does not fully align with the claimed goal in the introduction. If W1 and W2 can be addressed, I would likely consider raising my score.

---

> ### Author Response · Authors · 2024-11-20
> **Rebuttal Part 1**
>
> Thank you very much for your thorough and insightful review of our paper. We greatly appreciate your time, effort, and the valuable suggestions you have provided, and we are committed to making the necessary revisions to enhance the quality of our work. Below, we address your comments in detail.
>
> ## W1
>
> Thank you for this important feedback regarding our choice of correlation metrics. We have conducted a thorough re-analysis of our methodology and results:
>
> **1. Metric Selection and Justification**:
>
> - We agree that **Pearson correlation** is more appropriate for comparing evaluation metrics with human ratings, as it effectively **measures linear relationships across different rating methods and scales**.
> - While **Spearman correlation** was considered, it proved **unsuitable due to our discrete 5-point scale ([0,1,2,3,4])**, as rank-based analysis **would be compromised by numerous tied rankings**.
> - We **retained Krippendorff's Alpha** specifically for **human-human (H-H) and GPT-human (G-H)** reliability measurements in our case, as we used **identical evaluation criteria for both**. This dual use (for H-H and G-H) enables **direct comparisons with prior work like ImagenHub**.
>
> **2. Updated Analysis**: We have now computed Pearson correlations between all evaluation metrics and human ratings, providing a more rigorous quantitative comparison. This analysis will be added to the paper in final version:
>
> | Method | T2I Model | Concept Preservation | | | | Prompt Following | | |
> |---|---|---|---|---|---|---|---|---|
> | | | H-H | G-H | D-H | C-H | H-H | G-H | C-H |
> | Textual Inversion | SD v1.5 | 0.685 | 0.576±0.005 | 0.499±0.005 | 0.546±0.002 | 0.506 | 0.491±0.004 | 0.367±0.004 |
> | DreamBooth | SD v1.5 | 0.647 | 0.611±0.002 | 0.547±0.003 | 0.531±0.012 | 0.531 | 0.526±0.028 | 0.302±0.012 |
> | DreamBooth LoRA | SDXL v1.0 | 0.657 | 0.640±0.015 | 0.474±0.030 | 0.513±0.015 | 0.474 | 0.417±0.029 | 0.211±0.018 |
> | BLIP-Diffusion | SD v1.5 | 0.614 | 0.386±0.001 | 0.158±0.005 | 0.274±0.000 | 0.637 | 0.602±0.015 | 0.420±0.004 |
> | Emu2 | SDXL v1.0 | 0.745 | 0.733±0.002 | 0.721±0.011 | 0.701±0.013 | 0.445 | 0.436±0.009 | 0.308±0.012 |
> | IP-Adapter-Plus ViT-H | SDXL v1.0 | 0.603 | 0.407±0.022 | -0.043±0.010 | 0.132±0.001 | 0.579 | 0.581±0.010 | 0.334±0.011 |
> | IP-Adapter ViT-G | SDXL v1.0 | 0.591 | 0.464±0.007 | 0.060±0.024 | 0.165±0.027 | 0.511 | 0.583±0.015 | 0.325±0.034 |
> | Ratio | | 100% | 83.31% | 50.72% | 60.98% | 100% | 98.71% | 61.48% |
>
> **3. Strengthened Claims**: Our original conclusion that "Table 1 results show that DreamBench++ aligns better with humans than DINO or CLIP models..." is now supported by two complementary lines of evidence:
>
> - The new correlation analysis quantitatively demonstrates **stronger alignment between GPT and human evaluations**
> - This **aligns with experience expectations**, given that DreamBooth LoRA SDXL v1.0 is known to excel in Concept Preservation, unless some methods completely overfit the subject.
>
> **4. Data Transparency**: To facilitate reproducibility, we have:
>
> - Uploaded all rating data (GPT, DINO, CLIP, and human) to Google Drive
> - Updated our anonymous GitHub repository (https://github.com/dreambench/dreambench_plus) with relevant analysis (Krippendorff's Alpha and Pearson Correlation) code
> - Uploaded the original images generated by the method mentioned in the paper to Google Drive
> | Data |
> | --- |
> | [DreamBench++ Evaluation Dataset](https://drive.google.com/file/d/1z9NAugz-1FoGJGOPBxxb9zrDMxGa0ABs/view?usp=share_link) |
> | [Full Samples with 7 mehtods](https://drive.google.com/file/d/1cGz6EMRNpFXO8gRYuaeKglO6FlZWZU8V/view?usp=share_link) |
> | [human rating data](https://drive.google.com/file/d/1xAMRqi20qd-FqO3l0IgYreKnsESsjxeD/view?usp=share_link) |
> | [GPT rating data](https://drive.google.com/file/d/1u8r6djpNxOcMcW2IqDqwJX1zCDBQsVOf/view?usp=share_link) |
> | [DINO rating data](https://drive.google.com/file/d/1hcTtFsJfCMq1yZbCobiHciWcbOEBak2n/view?usp=share_link) |
> | [CLIP-I rating data](https://drive.google.com/file/d/1SwTWb1KfwzutLXPeWnuOkJ3RDkKgqRvz/view?usp=share_link) |
> | [CLIP-T rating data](https://drive.google.com/file/d/1dLSuVPayBoXuyCUqu-LEbxCQWMnzaMIl/view?usp=share_link) |
>
>
> We appreciate your feedback, which has helped us strengthen both our analysis and presentation of results. The final version will include these updates to provide a more comprehensive and rigorous evaluation framework.

---

> ### Author Response · Authors · 2024-11-20
> **Rebuttal Part2**
>
> ## W2
>
> Thank you for this insightful suggestion. We agree that a dedicated analysis of model performance across different generation types would strengthen our paper's alignment with its core objective of identifying superior technical approaches. We will incorporate the following comprehensive analysis in our revision:
>
> - **Concept Preservation Performance**
>   - **Animal:** Both fine-tuning and encoder-based personalization methods demonstrate strong performance in preserving animal concepts, primarily due to:
>     - Consistent visual features within species from a human cognition perspective
>     - For example, corgis exhibit highly unified visual representations, explaining their frequent use as test cases across personalization methods
>   - **Human:** Encoder-based methods significantly outperform fine-tuning approaches in human concept preservation, attributable to:
>     - Human observers' heightened sensitivity to facial feature accuracy
>     - Encoder-based methods have more precise capture and injection of subtle facial features
>     - Fine-tuning methods' tendency to focus on global image features rather than specific facial details
>     - This also confirms that current Identity-Preserving Generation (like InstantID) prefer encoder-based methods
>   - **Object:** The two approaches show complementary strengths
>     - Fine-tuning methods:
>       - Higher performance ceiling for common objects
>       - Better handling of objects with standardized appearances
>     - Encoder-based methods:
>       - Provide reliable baseline performance for rare/complex objects
>       - More consistent across diverse object types
>   - **Style:** Both approaches perform well in style preservation, likely because:
>     - Style features are more subjective and flexible
>     - Style transfer requires capturing high-level visual characteristics
>     - Both methods effectively learn and reproduce stylistic elements
> - **Prompt Following Capability**
>   - **Base Model Impact:** SDXL demonstrates superior prompt following compared to SD, driven by:
>     - More sophisticated and bigger text encoder architecture.
>     - Exposure to broader, more diverse training datasets and enhanced capacity for complex text-image relationships.
>   - **Methodological Differences:**
>     - Fine-tuning methods show clear advantages in prompt adherence due to:
>       - Preservation of original text-image conditional distribution
>       - Direct learning from image-text pairs
>       - Better maintenance of semantic alignment when properly regularized
>     - Encoder-based methods face challenges in prompt following due to:
>       - Potential disruption of learned text-image relationships during feature injection
>
> ## W3 & W4
> Thank you very much for your suggestions on the presentation of the paper. We will include more detailed table explanations and more intuitive leaderboard tables in the final version.
>
> ## Q1
>
> Our choice of a 0-4 scale was deliberate and grounded in both empirical evidence and practical considerations. While normalizing to [0,1] would **align with traditional metrics**, our **five-point scale (0,1,2,3,4) optimizes for human annotation reliability and score discriminability**. Through empirical validation, we found that this granularity **strikes an optimal balance** - it provides sufficient resolution to capture meaningful differences while ensuring that each score point maintains a clear, distinct meaning to annotators. When we experimented with finer-grained scales, even experienced human annotators struggled to consistently differentiate between adjacent scores, resulting in lower inter-rater reliability as measured by Krippendorff's Alpha.
>
> ## Q2
>
> The validity of GPT's automated evaluations is substantiated through multiple verification mechanisms. First, we established strict scoring criteria for human raters, which were then used to guide GPT's evaluations (As GPT’s Prompt). To rigorously verify GPT's performance, we employed **Krippendorff's Alpha to measure the agreement between GPT's scores and human ratings**, treating GPT as an additional rater within our evaluation framework. **The strong inter-rater agreement we observed indicates that GPT's reasoning and scoring closely align with human judgment patterns.** Additionally, we manually **examined the reasoning provided by GPT across different score levels** to ensure that the explanations consistently reflect the assigned scores and adhere to our predefined criteria. This dual verification approach - quantitative through Krippendorff's Alpha and qualitative through reasoning analysis - provides robust evidence for the reliability of our automated evaluation system.

---

> ### Author Response · Authors · 2024-11-20
> **Rebuttal Part 3**
>
> ## Q3
>
> The human annotation effort encompassed a comprehensive generated images of **18.9k samples (7 methods × 9 prompts × 150 images × 2 evaluations per sample)**. We employed **7 annotators** from a labeling company, each evaluating approximately 2.7k samples over a **two-week period with 8-hour workdays**. Compared to ImagenHub's three-level scoring system [0,1,2], we expanded to a five-level scoring system [0,1,2,3,4], providing more granular assessment while achieving higher annotation consistency. To ensure robust evaluation quality, we implemented a systematic annotation protocol:
>
> - **Scoring Criteria Development:** Paper authors (domain experts) developed detailed scoring standards based on generation results from 7 classic models. These standards were explicitly written into the GPT prompt, including specific Scoring Criteria and Scoring Range. We also selected typical examples (independent of the DreamBench++ dataset) as annotation references.
> - **Annotation Quality Control:** We developed a dedicated annotation platform and implemented multiple rounds of pilot annotations. To ensure genuine annotation quality, these pilot annotations were not pre-disclosed to annotators. Early quality checks revealed typical issues like over-reliance on middle values (2 points), tendency to choose extreme values, and fluctuating annotation focus. Through continuous quality feedback and training, annotation quality reached expected levels after 3-4 iterations.
> - **Formal Annotation and Monitoring:** During the formal annotation phase, we used a continuous quality monitoring mechanism. Quality assessment was conducted after each round, with decisions on additional training or proceeding to the next round based on results. This iterative feedback mechanism ensured overall annotation quality stability.
> - **Consistency Verification:** The final human rating consistency significantly outperformed ImagenHub, validating the effectiveness of our annotation strategy. This achievement stems from a rigorous annotation process and annotators' professional commitment.
>
> We invested substantial resources to ensure data quality, hoping this dataset will serve the broader research community, whether for model evaluation or training specialized evaluation models.
>
> ## Q4
> We also appreciate your suggestions, and we will evaluate more methods to support our conclusions in the final version.

---

> ### Comment · Reviewer_RNsM · 2024-11-21
> **Response to rebuttal.**
>
> Thanks for the detailed response. I have adjusted the score accordingly (3 -> 6). Would love to see the continuous effort in maintaining the bench.

---

> > ### Author Response · Authors · 2024-11-21
> > **Thank you!**
> >
> > Dear Reviewer RNsM,
> >
> > Thank you for your acknowledgement of our efforts! We’re also grateful for your score adjustment! This encourages us to continue refining and maintaining the bench to meet the community's expectations.
> >
> > Once again, thank you for your invaluable support and insights.
> >
> > Best regards,
> > Authors

---

### Official Review · Reviewer_Xmsj · 2024-11-02

**Soundness:** 3
**Presentation:** 3
**Contribution:** 3
**Rating:** 6
**Confidence:** 3

**Summary:**

The paper presents DREAMBENCH++, a human-aligned benchmark for evaluating personalized image generation models using advanced multimodal models like GPT-4o for automated, human-like assessments. It addresses the misalignment and cost issues of traditional evaluation methods by focusing on prompt following and concept preservation with a diverse dataset. DREAMBENCH++ demonstrates superior alignment with human evaluations, offering a comprehensive and unbiased framework to advance personalized image generation research.

**Strengths:**

This paper proposes a metric that closely aligns with human preferences for evaluating personalized generated images and introduces a suitable benchmark. Compared to existing benchmarks, it constructs a much more diverse and extensive dataset and demonstrates evaluation results that are better aligned with humans than CLIP and DINO.

**Weaknesses:**

- For text-to-image generation models, there are many metrics, including TIFA[1], that improve upon CLIP. It is necessary to demonstrate alignment with humans not only compared to CLIP and DINO but also in comparison with these existing studies.

[1] Hu, Yushi, et al. "Tifa: Accurate and interpretable text-to-image faithfulness evaluation with question answering." Proceedings of the IEEE/CVF International Conference on Computer Vision. 2023.

- The prompts used for GPT-4o are abstract. Studies that evaluate generated images using VQA typically ask absolute questions, such as the presence, location, or number of objects, to fully leverage GPT-4o’s performance. However, the prompting approach in this paper treats GPT-4o like a human, and the justification relies only on citations from other studies. In practice, some degree of hallucination occurs when GPT-4o evaluates images, which the paper overlooks, making this a significant drawback.

-The scoring system from 0 to 4 is ambiguous. Some images may have around 10 objects, with 1 or 2 that do not match, while others may have all objects correct but differ in style. What are their scores? While GPT-4o might provide consistent evaluations, humans are less likely to be consistent. To address this, a larger user study should have been conducted, but only seven people participated, with each instance evaluated by at least two annotators. This means some images were reviewed by just two people, making the number of annotators insufficient.

**Questions:**

Please see the weakness part. In addition,

- Why is the comparison scheme considered unsuitable in lines 198-199 if the scoring results are sensitive? Was "comparing" mistakenly written as "scoring"?

- Stable Diffusion often generates noisy backgrounds effectively and frequently. Why did the authors judge it as unsuitable for personalized generation and remove it?

- Are the example prompts shown in Figure 4 intended for personalized editing with those specific prompts?

- In Table 1, the highest-scoring model is consistent across Human, GPT, DINO-I, CLIP-I, and CLIP-T. Can DREAMBENCH++ really be said to be more aligned with humans than DINO or CLIP?

- Contrary to the authors' claim, Figure 6 does not clearly show that DINO or CLIP prefer shape and overall styles. Also, do the authors believe that favoring shape and overall styles is not aligned with human preference?

---

> ### Author Response · Authors · 2024-11-24
> **Rebuttal Part 1**
>
> We thank you for your acknowledgement of our work and appreciate your suggestions. We are committed to answering your concerns in detail and making the necessary revisions to enhance the quality of our work.
>
> ## Weakness 1: Comparison with newer metrics such as TIFA
>
> Thank you for your suggestion and reference. The need of comparing our work with existing text-to-image (T2I) benchmarks has also been raised by other reviewers. We will include a more comprehensive experiment that compares our method with existing T2I benchmarks in the appendix of the final version. Here, we would like to highlight some of the key dinstinctions between our benchmark and other existing benchmarks.
>
> - **Designed for Different Tasks.** Existing benchmarks such as TIFA are mostly designed to evaluate **text-to-image (T2I) generation tasks**, i.e., generating an image based on text description. In contrast, our proposed DreamBench++ focuses on the **personalized generation task**, a more complex challenge that requires generating an image based on not only text instructions but also a reference image (e.g., given a reference picture of my dog, generate a picture of my dog on the moon).
> - **Different Criteria.** The core criterion of T2I benchmarks is **fine-grained text adherence**, i.e., assessing whether the generated image accurately reflects detailed textual instructions. Additionally, our DreamBench++ not only focuses on text adherence but also **visual** **concept preservation** (e.g., assessing how well the dog in the generated image preserves the unique characteristics of the dog depicted in the reference picture).
> - **Different** **Methodology****.** TIFA evaluates generated images by posing a series of VQA-based questions. For instance, given the text description *"A person sitting on a horse in air over gate in grass with people and trees in background,"* TIFA might ask questions such as “What is the animal?”, “Is there a gate?”, or “Is the horse in the air?”. The answers are then used to compute a fine-grained score.
>
> **Predictably:**
>
> - In terms of **prompt following ability** (text adherence), our method may not perform as well as benchmarks specifically designed for T2I tasks. This is because:
>   - Our design does not emphasize fine-grained text adherence;
>   - Instead, we use a scoring mechanism similar to that for concept preservation, where GPT determines scores based on provided criteria and ranges.
> - Nevertheless, even with this **relatively simple design**, the correlation between GPT’s scores and human evaluations consistently exceeds 90% (noting that we will update Table 4 in response to Reviewer RNsM's feedback by replacing Krippendorff’s alpha with Pearson Correlation Coefficient, with the results remaining unchanged).

---

> ### Author Response · Authors · 2024-11-24
> **Rebuttal Part 2**
>
> ## Weakness 2 & 3: Possible hallucination of GPT-4o under subjective criteria
>
> ### **On the Use of Abstract Prompts with GPT-4o**
>
> We acknowledge the reviewer's observation that VLMs perform better with objective and specific queries (e.g., questions about object presence, location, or count). **However, the personalized image generation task inherently involves subjective and abstract criteria.** For example, determining whether a generated dog matches the reference dog goes beyond simple comparisons of size, color, or position. It often involves **assessing nuanced and complex semantic relationships**, such as whether the generated dog “is the same dog” as in the reference image. Similarly, textual instructions may contain a variety of intents, including altering specific attributes of objects while preserving others.
>
> Our approach to GPT-4o scoring is **justified primarily by its alignment with human evaluations, as evidenced by the high Krippendorff’s alpha between GPT-4o scores and human annotations**. While GPT-4o operates as a “black box” to some extent, **it provides justifications for its scores, which exhibit strong similarity to human reasoning.** This consistency supports the feasibility of using **GPT-4o as a proxy for human evaluation** in this context.
>
> ### **On the Scoring System (0-4 Scale)**
>
> The choice of a 0-4 scale was carefully considered, balancing practical and empirical factors. While normalizing scores to [0,1] might align with conventional metrics, we found through empirical validation that a five-point scale (0, 1, 2, 3, 4) offers better granularity for capturing meaningful differences in image quality. **This scale ensures clear and distinct score meanings for annotators, optimizing annotation reliability and discriminability.**
>
> **When testing finer-grained scales, we observed a significant drop in inter-rater reliability, as even experienced annotators struggled to consistently differentiate between adjacent scores.** The five-point scale mitigated these issues, resulting in higher Krippendorff’s alpha values and improved consistency.
>
> #### On the Number of Human Annotators and Annotation Protocol
>
> The human annotation process was extensive and rigorous, covering 18.9k generated samples (7 methods × 9 prompts × 150 images × 2 evaluations per image). Seven professional annotators evaluated approximately 2.7k samples each over a two-week period with 8-hour workdays.
>
> **To address concerns about annotation consistency and quality, we implemented a robust protocol:**
>
> 1. **Scoring Criteria Development**: The paper authors, as domain experts, developed explicit and detailed scoring standards based on results from seven classic models. These standards were incorporated into the GPT prompt, including specific scoring criteria and example-based references to ensure clarity.
> 2. **Quality Control**: Multiple rounds of pilot annotations were conducted to identify and address common issues, such as over-reliance on middle scores, preference for extremes, and inconsistent focus. Iterative feedback and training improved annotator performance to the desired level within 3-4 iterations.
> 3. **Formal Annotation**: Continuous monitoring during formal annotations ensured quality stability. Each round was followed by quality assessments, with decisions made regarding further training or progression based on performance metrics.
> 4. **Consistency Verification**: Final human rating consistency significantly surpassed benchmarks like ImagenHub, validating the effectiveness of our annotation process.
>
> Our resource-intensive efforts underscore our commitment to creating a high-quality dataset that can serve the research community for both model evaluation and as a resource for training evaluation-specific models.
>
> We acknowledge that a larger number of annotators could provide additional robustness. However, **we emphasize that our protocol and iterative training ensured that the final annotation quality and consistency exceeded existing benchmarks**. This focused approach allowed us to maintain a **balance between resource constraints and dataset quality**, ensuring that every annotator adhered to rigorous standards.

---

> ### Author Response · Authors · 2024-11-24
> **Rebuttal Part 3**
>
> Q1: Yes, "comparing" was mistakenly written as "scoring". Thank you for pointing that out.
>
> Q2: I'm a bit unclear about what the 'Why did the authors judge it as unsuitable for personalized generation and remove it?' means; can you explain it in more detail?
>
> Q3: **Regarding the example prompts shown in Figure 4**: Yes, we generated 9 different prompts for each reference image using GPT for image editing (personalization). These 9 prompts vary in difficulty.
>
> Q4: **About the model scoring in Table 1**: Table 1 does not clearly reflect the consistency between GPT and human evaluation; **it should be combined with Table 4 for better understanding, and we will supplement the explanation in the final version.**
> After discussions with reviewer RNsM, we realized that Krippendorff’s alpha is not a suitable metric for evaluating the consistency between DINO/CLIP and human evaluations. The reason is that the scoring criteria of DINO and CLIP differ significantly from those of humans, while Krippendorff’s alpha assumes consistent criteria across raters. **To better measure consistency, the most appropriate metric is the Pearson correlation coefficient**, which directly reflects the linear correlation between the model scores and human scores. Based on the above insights, we recalculated the relevant metrics using the Pearson correlation coefficient and updated the corresponding table in our paper. Despite this change, the main conclusions of our study remain consistent, further demonstrating the robustness and validity of our method.
>
> We have now computed **Pearson correlations between all evaluation metrics and human ratings**, providing a more rigorous quantitative comparison. This analysis will be added to the paper in final version:
>
> | Method                | T2I Model | Concept Preservation |             |              |             | Prompt Following |             |             |
> | --------------------- | --------- | -------------------- | ----------- | ------------ | ----------- | ---------------- | ----------- | ----------- |
> |                       |           | H-H                  | G-H         | D-H          | C-H         | H-H              | G-H         | C-H         |
> | Textual Inversion     | SD v1.5   | 0.685                | 0.576±0.005 | 0.499±0.005  | 0.546±0.002 | 0.506            | 0.491±0.004 | 0.367±0.004 |
> | DreamBooth            | SD v1.5   | 0.647                | 0.611±0.002 | 0.547±0.003  | 0.531±0.012 | 0.531            | 0.526±0.028 | 0.302±0.012 |
> | DreamBooth LoRA       | SDXL v1.0 | 0.657                | 0.640±0.015 | 0.474±0.030  | 0.513±0.015 | 0.474            | 0.417±0.029 | 0.211±0.018 |
> | BLIP-Diffusion        | SD v1.5   | 0.614                | 0.386±0.001 | 0.158±0.005  | 0.274±0.000 | 0.637            | 0.602±0.015 | 0.420±0.004 |
> | Emu2                  | SDXL v1.0 | 0.745                | 0.733±0.002 | 0.721±0.011  | 0.701±0.013 | 0.445            | 0.436±0.009 | 0.308±0.012 |
> | IP-Adapter-Plus ViT-H | SDXL v1.0 | 0.603                | 0.407±0.022 | -0.043±0.010 | 0.132±0.001 | 0.579            | 0.581±0.010 | 0.334±0.011 |
> | IP-Adapter ViT-G      | SDXL v1.0 | 0.591                | 0.464±0.007 | 0.060±0.024  | 0.165±0.027 | 0.511            | 0.583±0.015 | 0.325±0.034 |
> | Ratio                 |           | 100%                 | 83.31%      | 50.72%       | 60.98%      | 100%             | 98.71%      | 61.48%      |
>
> Q5: **Regarding the claim in Figure 6**: What we want to express is that many high-scoring cases selected by DINO maintain the same outline and posture (for example, the posture of the dog in the reference image and the generated image is the same, and both only show the upper body, but the facial details and colors are different; humans are more likely to think these are not the same dog, similar to the case of basketball players). Although the high-scoring cases from GPT may have many changes in posture/outlines and backgrounds, they exhibit stronger ID consistency.

---

> ### Author Response · Authors · 2024-11-24
> **Looking forward to your feedback**
>
> Dear Reviewer Xmsj,
>
> Thank you for your great reviewing efforts and your supportive assessment of our work! With the discussion period drawing to a close, we expect your feedback and thoughts on our reply.
>
> We look forward to hearing from you, and we can further address unclear explanations and remaining concerns if any.
>
> Best,\
> Authors

---

> > ### Comment · Reviewer_Xmsj · 2024-11-27
> >
> > Authors,
> >
> > Thank you for the detailed rebuttal and explanation.
> > - Additional benchmark comparison: I agree that personalization is a different task compared to text-to-image translation.
> > - Q2: I meant by "why background should not be considered for evaluation?". Background sometimes can be a important part of image, highly related to the foreground.
> > - Thanks for the report regarding Pearson correlations.
> > - Remaining answers are convinced.
> >
> > Considering all, I will keep my original rating, 6.

---

> > > ### Author Response · Authors · 2024-12-01
> > > **Thank you!**
> > >
> > > Dear Reviewer Xmsj,
> > >
> > > Thank you for your thoughtful and detailed feedback on our submission. We truly appreciate the time and effort you put into reviewing our work. Your comments have provided valuable insights and have helped us to further clarify and refine our approach.
> > >
> > > Best, Authors

---

### Official Review · Reviewer_Rkya · 2024-11-03

**Soundness:** 2
**Presentation:** 3
**Contribution:** 2
**Rating:** 6
**Confidence:** 3

**Summary:**

This work introduces a new benchmark for personalized text-to-image generation, including a dataset with a greater number of images and prompts compared to DreamBench, and an automatic evaluation method that assesses two key aspects of the task: (i) concept preservation and (ii) prompt following, using multimodal large language models like GPT. This new evaluation method addresses the limitations of previous DINO and CLIP and achieves higher human alignment.

**Strengths:**

It fills a gap in human-aligned and automated evaluation of the personalized text-to-image generation task by introducing LLMs. The dataset and evaluation metric is clearly described in detail.

**Weaknesses:**

1. The new dataset aims to include a larger variety of images but the source of these images is still relatively narrow (only 3 websites). The new dataset can still be biased due to the preference of these websites, and there is no sufficient evidence to show its diversity (only the visualization of t-SNE).
2. It is claimed that the method is transferable to other foundation models. It is not straightforward because the method is specifically designed for GPT4 and no experiment showed its transferability.
3. The dataset only contains 1 image for every instance. Although multiple images are claimed to be unnecessary, the results in Fig. 9 show serious overfitting when using only 1 reference image.
4. There are other key aspects that need evaluation. For example, a common problem of fine-tuning-based methods is overfitting. We generally don't want the generated images to be too similar to the reference image except for the identity of the object. It is worth trying to evaluate the overfitting problem using GPT.

**Questions:**

How are the numbers calculated? Can you specify how you used Krippendorff’s alpha value? And where does the number of 54.1% and 50.7% come from? I noticed that the CLIP and DINO scores fairly correlate with human scores.

---

> ### Author Response · Authors · 2024-11-23
> **Rebuttal Part 1**
>
> Thank you very much for your thorough and insightful review of our paper. We greatly appreciate your time, effort, and the valuable suggestions you have provided, and we are committed to making the necessary revisions to enhance the quality of our work. Below, we address your comments in detail.
> ## W1
>
> Thank you for the your insightful comments regarding the **dataset’s sources and diversity**. We address these concerns as follows:
>
> First, **regarding the data sources**, our dataset is not strictly limited to three websites. **Google Images is a comprehensive search engine that accesses images across the entire web.** The primary reason for selecting these three sources lies in their **copyright transparency and suitability for academic use**. Ensuring proper copyright compliance is critical for constructing a dataset, as it **avoids potential issues that have led to the removal of some datasets**. Furthermore, these sources provide high-quality images with sufficient diversity to meet our needs.
>
> Second, **regarding dataset diversity**, our goal was to **maximize diversity with a relatively small number of images**, considering the **cost-intensive nature of GPT-based evaluations.** To achieve this, we designed a **rigorous construction pipeline** (illustrated in Fig. 4) that ensures diversity at multiple stages:
>
> 1. **Keyword Generation**: We started by categorizing personalized tasks into broad groups (e.g., animals, humans, styles, and objects) and used LLMs to generate a large number of keywords. Additionally, we leveraged the top 200 keywords from the Unsplash dataset and included numerous human-proposed keywords. This step ensures semantic diversity in the dataset’s sources.
> 2. **Data Retrieval**: Using Google Images, we retrieved images from across the web, inherently ensuring diversity due to the wide range of available content.
> 3. **Manual Filtering**: After data retrieval, we conducted manual filtering to remove images with highly repetitive styles or categories, further enhancing the dataset’s diversity and quality.
>
> While our diversity evaluation primarily relies on t-SNE visualization, the results clearly demonstrate broader data coverage and reduced clustering. Furthermore, every step in our pipeline is explicitly designed to prioritize diversity. **We acknowledge that there is room for improvement in quantifying diversity, and we plan to explore more comprehensive evaluation methods in future work.**
>
> In conclusion, despite the limited sources and simple evaluation methods, we believe our dataset demonstrates significant advantages in diversity and robustness, supporting future personalized tasks effectively.

---

> ### Author Response · Authors · 2024-11-23
> **Rebuttal Part 2**
>
> ## W2
>
> Thank you for raising this valuable question. We agree that validating the transferability of our method is an important scientific concern. To address this, we conducted additional tests on various multimodal foundation models and provided comparative results in our experiments.
>
> In the last column of the table, the *ratio* (i.e., the average proportion of H-H consistency) quantifies the alignment between model evaluations and human evaluations, reflecting overall performance. The experimental results indicate the following:
>
> 1. **Performance of the GPT-4o series**: The capabilities of GPT-4o are **continuously evolving**. We observed that the 2024-08-06 version achieved higher human-machine consistency compared to the 2024-05-13 version. Furthermore, the mini version (GPT-4o-mini-2024-07-18), despite its smaller size, also achieved notable results.
> 2. **Performance of Claude 3.5**: We also tested Claude-3.5-sonnet-20241022, which is considered to have capabilities comparable to GPT-4o. **This model exhibited a similar level of consistency ratio, further supporting the transferability of our method.**
> 3. **Performance of Gemini**: Due to **API availability limitations**, we could only test an **earlier version of Gemini-1.5-pro-001**, which performed relatively weaker. The newer versions of the model may demonstrate improved consistency.
>
> Overall, our results indicate that the proposed method is not only applicable to GPT-4o but also generalizes well to other advanced multimodal models. This cross-model consistency enhances the practical usability and scientific value of the method. Thank you for helping us refine this argument through your insightful question.
>
> | Method                     | TI SD   | DreamBooth | DreamBooth-L | BLIP-D  | Emu2      | IP-Adapt.-P | IP-Adapt. | Ratio  |
> | -------------------------- | ------- | ---------- | ------------ | ------- | --------- | ----------- | --------- | ------ |
> | T2I Model                  | SD v1.5 | SD v1.5    | SDXL v1.0    | SD v1.5 | SDXL v1.0 | SDXL v1.0   | SDXL v1.0 |        |
> | H-H (as ground truth)      | 0.685   | 0.647      | 0.656        | 0.613   | 0.746     | 0.602       | 0.591     | 100%   |
> | gpt-4o-2024-05-13 (paper)  | 0.544   | 0.596      | 0.641        | 0.362   | 0.669     | 0.366       | 0.458     | 79.47% |
> | gpt-4o-2024-08-06          | 0.558   | 0.625      | 0.645        | 0.383   | 0.708     | 0.427       | 0.502     | 84.23% |
> | gpt-4o-mini-2024-07-18     | 0.496   | 0.575      | 0.538        | 0.274   | 0.702     | 0.238       | 0.289     | 67.22% |
> | claude-3-5-sonnet-20241022 | 0.500   | 0.583      | 0.623        | 0.354   | 0.625     | 0.278       | 0.427     | 74.00% |
> | gemini-1.5-pro-001         | 0.487   | 0.547      | 0.514        | 0.267   | 0.653     | 0.253       | 0.202     | 63.04% |

---

> ### Author Response · Authors · 2024-11-23
> **Rebuttal Part3**
>
> ## W3
>
> Regarding the concern about our benchmark using only a single image per instance, we appreciate the reviewers' perspective and would like to clarify our rationale for this design choice.
>
> - **Practical considerations**: In many practical applications, especially in scenarios where personalization is required, the availability of multiple reference images for each instance is often limited. Therefore, we designed our benchmark to assess the capability of personalization using a single image, which we believe reflects real-world constraints.
> - **We primarily focus on dataset diversity**: While we highly appreciate Benchmarks that contains multiple images like DreamBooth and CustomConept101, our work adopts a slightly different focus. Our primary emphasis is on ensuring the diversity of the dataset in terms of styles and types of objects. We believe this approach facilitates a more comprehensive and balanced comparison across various personalized image generation methods.
> - **Performance are sufficient with single images**: We have compared generation qualities of generation results on DreamBench by using multiple images and single image per instance, and we found that tested methods such as SDXL can achieve their full capabilities even with just a single image. Therefore we believe that our benchmark is sufficient in evaluating the full capabilities of these methods within the scope of our research.
> - **Difficulty in constructing datasets of intance with multiple images**: Collecting multiple diverse images for the same instance is challenging, and in our existing personalization dataset, the diversity of instances with multiple images is not high. We lacked an effective way to address this issue at the time, and ignored the excellent CustomConcept101 dataset at the time, so we did not consider constructing a dataset of instance with multiple images. In the final version, we will expand our dataset based on CustomConcept101 to improve the diversity, fairness, and robustness of our evaluation framework.
> - **We leave this for future research**: We fully concur with the reviewers that collecting multiple diverse images for a single instance is indeed a valuable approach. While we respectfully maintain that utilizing a single image per instance does not diminish the significance of our benchmark or the validity of our main results, we deeply appreciate the reviewers' suggestion in this regard and recognize this as an excellent direction for future research. We will discuss the potential for expanding our dataset in this manner in the final version of the paper.

---

> ### Author Response · Authors · 2024-11-23
> **Rebuttal Part 4**
>
> ## W4
>
> Thank you for pointing this out. In fact, this was **one of the core motivations behind constructing our benchmark**. Through our exploration of personalized generation, we identified significant limitations in the current mainstream metrics (e.g., DINO and CLIP), particularly their **inability to accurately distinguish between overfitting and true generative quality when evaluating models’ personalization capabilities**.
>
> Specifically, DINO and CLIP metrics often place excessive emphasis on the **visual similarity between the generated and reference images**. Many works **achieve high scores by overfitting to the reference image**, where the generated subject is nearly identical to the reference. However, such results fail to reflect true personalization capabilities. In contrast, DreamBooth, when appropriately fine-tuned, can generate images that align well with the text description while retaining the subject's identity. **Although this approach may lead to lower DINO/CLIP scores, it demonstrates better personalization, as the subject in the generated images exhibits meaningful variations in actions, poses, expressions, and appearances.**
>
> To address this issue, we devised a more interpretable and discriminative evaluation method by combining two metrics: Concept Preservation (CP) and Prompt Following (PF). By calculating the product **CP·PF, we can better quantify the overall performance of a model in personalized generation** while avoiding biases from single metrics. Additionally, we observed that the **CP/PF ratio can provide insights into overfitting**: higher ratios often indicate cases where the generated images excessively mimic the reference image at the expense of faithfully following the text description.
>
> In the final version of the paper, we plan to include a more detailed table and explanation in the appendix to clearly present the performance of different models. Below is a summary of our experimental results:
>
> |                       | T2I Model | CP    | PF    | CP*PF | CP/PF |
> | --------------------- | --------- | ----- | ----- | ----- | ----- |
> | DreamBooth LoRA       | SDXL v1.0 | 0.598 | 0.865 | 0.517 | 0.69  |
> | IP-Adapter ViT-G      | SDXL v1.0 | 0.593 | 0.640 | 0.380 | 0.93  |
> | Emu2                  | SDXL v1.0 | 0.528 | 0.690 | 0.364 | 0.77  |
> | DreamBooth            | SD v1.5   | 0.494 | 0.721 | 0.356 | 0.69  |
> | IP-Adapter-Plus ViT-H | SDXL v1.0 | 0.833 | 0.413 | 0.344 | 2.02  |
> | BLIP-Diffusion        | SD v1.5   | 0.547 | 0.495 | 0.271 | 1.11  |
> | Textual Inversion     | SD v1.5   | 0.378 | 0.624 | 0.236 | 0.61  |
>
> ## Q
> Thank you for your careful review and insightful questions. Below are our detailed responses:
>
> 1. **On the calculation and rationale for using Krippendorff’s alpha** We calculated Krippendorff’s alpha using the implementation provided by the open-source repository [fast-krippendorff](https://github.com/pln-fing-udelar/fast-krippendorff). We chose this metric because it is widely regarded as **the most flexible and robust measure for evaluating inter-rater reliability**. It effectively handles missing data and supports various data types, including ordinal and continuous data. Furthermore, **Krippendorff’s alpha has been widely adopted in related studies, such as GPT4Eval3D and ImageHub.** This makes our choice both theoretically sound and conducive to comparisons with results from these works.
> 2. **On the origin of the numbers 54.1% and 50.7%** The numbers 54.1% and 50.7% were calculated as follows: 1) 54.1%=79.64%−25.54%; 2)50.7%=93.18%−42.48%. Here, 79.64% and 93.18% represent the agreement scores between GPT and human evaluations, while 25.54% and 42.48% represent the agreement scores between DINO and human evaluations. These differences quantify the relative improvement in alignment of GPT over DINO when compared to human evaluations.
> 3. **On the appropriateness of using Krippendorff’s alpha for DINO/CLIP consistency with humans** After discussions with **reviewer RNsM**, we realized that **Krippendorff’s alpha is not a suitable metric for evaluating the consistency between DINO/CLIP and human evaluations.** The reason is that the scoring criteria of DINO and CLIP differ significantly from those of humans, while Krippendorff’s alpha assumes consistent criteria across raters. To better measure consistency, the most appropriate metric is the **Pearson correlation coefficient**, which directly reflects the **linear correlation between the model scores and human scores**. Based on the above insights, we recalculated the relevant metrics using the Pearson correlation coefficient and updated the corresponding table in our paper. Despite this change, the main conclusions of our study remain consistent, further demonstrating the robustness and validity of our method.

---

> > ### Author Response · Authors · 2024-11-23
> > **Updated Pearson Analysis**
> >
> > **Updated Pearson Analysis**: We have now computed Pearson correlations between all evaluation metrics and human ratings, providing a more rigorous quantitative comparison. This analysis will be added to the paper in final version:
> >
> > | Method                | T2I Model | Concept Preservation |             |              |             | Prompt Following |             |             |
> > | --------------------- | --------- | -------------------- | ----------- | ------------ | ----------- | ---------------- | ----------- | ----------- |
> > |                       |           | H-H                  | G-H         | D-H          | C-H         | H-H              | G-H         | C-H         |
> > | Textual Inversion     | SD v1.5   | 0.685                | 0.576±0.005 | 0.499±0.005  | 0.546±0.002 | 0.506            | 0.491±0.004 | 0.367±0.004 |
> > | DreamBooth            | SD v1.5   | 0.647                | 0.611±0.002 | 0.547±0.003  | 0.531±0.012 | 0.531            | 0.526±0.028 | 0.302±0.012 |
> > | DreamBooth LoRA       | SDXL v1.0 | 0.657                | 0.640±0.015 | 0.474±0.030  | 0.513±0.015 | 0.474            | 0.417±0.029 | 0.211±0.018 |
> > | BLIP-Diffusion        | SD v1.5   | 0.614                | 0.386±0.001 | 0.158±0.005  | 0.274±0.000 | 0.637            | 0.602±0.015 | 0.420±0.004 |
> > | Emu2                  | SDXL v1.0 | 0.745                | 0.733±0.002 | 0.721±0.011  | 0.701±0.013 | 0.445            | 0.436±0.009 | 0.308±0.012 |
> > | IP-Adapter-Plus ViT-H | SDXL v1.0 | 0.603                | 0.407±0.022 | -0.043±0.010 | 0.132±0.001 | 0.579            | 0.581±0.010 | 0.334±0.011 |
> > | IP-Adapter ViT-G      | SDXL v1.0 | 0.591                | 0.464±0.007 | 0.060±0.024  | 0.165±0.027 | 0.511            | 0.583±0.015 | 0.325±0.034 |
> > | Ratio                 |           | 100%                 | 83.31%      | 50.72%       | 60.98%      | 100%             | 98.71%      | 61.48%      |

---

> > ### Comment · Reviewer_Rkya · 2024-11-24
> >
> > Thank you for your detailed and thoughtful rebuttal.

---

> > > ### Author Response · Authors · 2024-11-24
> > > **Thank you!**
> > >
> > > Dear Reviewer Rkya,
> > >
> > > Thank you for acknowledging our rebuttal and for your great efforts in reviewing this paper! We put a significant effort into our response, which includes several new experiments and discussions. We sincerely hope you can consider all these efforts for a re-assessment of the paper's rating.
> > >
> > > We look forward to hearing from you, and we can further address unclear explanations and remaining concerns, if any.
> > >
> > > Best,\
> > > Authors

---

> > > > ### Comment · Reviewer_Rkya · 2024-11-30
> > > >
> > > > I have adjusted my rating to 6. Look forward to seeing you expanding your benchmark to multiple images per instance in your future work.

---

> > > > > ### Author Response · Authors · 2024-12-01
> > > > > **Thank you!**
> > > > >
> > > > > Dear Reviewer Rkya,
> > > > >
> > > > > Thank you for your valuable feedback and suggestions. We will continue to work on expanding our benchmark to incorporate multiple images per instance in future work.
> > > > >
> > > > > Best,
> > > > > Authors

---

### Official Review · Reviewer_w3GE · 2024-11-03

**Soundness:** 3
**Presentation:** 3
**Contribution:** 3
**Rating:** 6
**Confidence:** 4

**Summary:**

This paper proposed a new T2I evaluation benchmark, DREAMBENCH++, which is introduced as a human-aligned benchmark for personalized image generation, addressing the limitations of current evaluations that are either misaligned with human judgment or time-consuming and expensive. The researchers systematically design prompts to make GPT models both human-aligned and self-aligned, enhanced with task reinforcement, and construct a comprehensive dataset of diverse images and prompts. By benchmarking 7 modern generative models, this paper demonstrates that DREAMBENCH++ achieves significantly more human-aligned evaluation, contributing to innovative findings in the field.

**Strengths:**

I appreciate that the authors engage a lot of effort to build a new benchmark, including design prompt, collecting images, and conducting experiments among several existing T2I models. The paper is written well and clear, the figure is informative, which is good.

**Weaknesses:**

1. I am confused about the motivation of building this new benchmark actually. As stated on line 084-085, "can we comprehensively evaluate these models to figure out which technical route is superior and where to head?", it's hard for reviewers to understand what distinguishes your benchmark compared with other existing benchmarks. Does DREAMBENCH++ assess more skills than existing T2I benchmarks? Does DREAMBENCH++ include more samples or higher quality images? A comparison table (DREAMBENCH++ v.s. existing benchmarks) is necessary to convince reviewers how the new benchmark can benefit T2I community.

2. Nowadays, there are several existing T2I benchmarks, for example, DALL-EVAL [1] and HRS-Bench [2]. So a new benchmark alone may not contribute enough in this field. It would be good to contribute one technical contribution to somehow address the challenge which is emphasized in the newly built benchmark.

[1] Cho, J., Zala, A., & Bansal, M. (2023). Dall-eval: Probing the reasoning skills and social biases of text-to-image generation models. In Proceedings of the IEEE/CVF International Conference on Computer Vision (pp. 3043-3054).
[2] Bakr, E. M., Sun, P., Shen, X., Khan, F. F., Li, L. E., & Elhoseiny, M. (2023). Hrs-bench: Holistic, reliable and scalable benchmark for text-to-image models. In Proceedings of the IEEE/CVF International Conference on Computer Vision (pp. 20041-20053).

**Questions:**

1. Can you do specific comparisons between DREAMBENCH++ and existing benchmarks like DALL-EVAL and HRS-Bench. For example, a comparison table showing the number of samples, types of skills assessed, and evaluation metrics used in each benchmark is necessary. This would help clarify the unique contributions of DREAMBENCH++.

2. May I ask if the authors have considered developing new evaluation metrics, proposing improvements to existing personalized T2I models based on benchmark insights?

---

> ### Author Response · Authors · 2024-11-20
>
> Thank you very much for your thorough and insightful review of our paper. We greatly appreciate your time, effort, and the valuable suggestions you have provided, and we are committed to making the necessary revisions to enhance the quality of our work.
>
> Below, we clarify our contributions as well as the distinction between our work and existing benchmarks in response to your concerns.
>
> ## 1. Our Motivation of Creating DreamBench++
>
> **Existing T2I benchmarks** (e.g., HRS-Bench and DALL-EVAL) focus on the **text-to-image generation task**, the core criteria of which is **fine-grained text adherence**, i.e. whether the generated image correctly reflects the detailed requirements in text instructions (e.g., generating objects in specific quantities or generate compositional objects). Some benchmarks also includes evaluation on **image quality**, and **potential biases** (e.g., regarding race and gender). For instance, HRS-Bench highlights strict adherence to textual instructions, such as generating images involving specific quantities or logical compositions (e.g., a cat chasing a dog), and evaluates models across multiple dimensions, including image quality, fairness, and social bias.
>
> In contrast, the core motivation of **DreamBench++** is to assess the **personalized generation task**, which is a more challenging task that generates an image based on **not only text instruction but also a reference image (e.g. generate a picture of my dog on the moon)**. In addition to the criteria mentioned above, the evaluation of personalized generation tasks also involves measuring the **similarity between the subject in the generated image and the subject in the reference image**. This task is distinct from traditional text-to-image evaluation methods and aims to address the lack of effective evaluation for personalized generation capabilities in existing benchmarks.
>
> ## 2. Limitations of Existing Evaluation Methods
>
> We notice that while existing evaluation methods are good at evaluating image-text alignment, they do not perform well in assessing the subject similarity between images, as exemplified in Figure 6. The most intuitive and accurate evaluation approach would be to manually assess the qualities of generated personalized images, i.e. conducting user studies. However, **current user studies suffer from notable limitations**:
>
> - **Lack of unified evaluation standards**: Different papers often adopt inconsistent evaluation metrics and processes, making the results difficult to compare.
> - **Lack of transparency**: Specific evaluation details are rarely disclosed, making it challenging to reproduce or verify conclusions objectively.
>
> As a widely adopted proxy to human evaluation, **traditional automated metrics (e.g., DINO Score and CLIP Score) are insufficient in reflecting the subject-level similarity between the generated and reference images**. As shown in Table 4, these metrics exhibit low correlation with human subjective evaluations. (Note that we would have updated Table 4 following Reviewer RNsM's feedback to replace Krippendorff's alpha metric with Pearson Correlation Coefficient, the outcome stays the same)
>
> ## 3. Technical Innovation
>
> To address these complex challenges, we propose a **universal and robust evaluation framework for personalized generation**, leveraging multimodal large models to quantify subjective similarity. The core innovations of this approach include:
>
> - **Shifting from comparison to scoring tasks**: Initially, we attempted a comparison task where the multimodal model judged which of two generated images was closer to the reference image. However, experiments revealed severe **positional bias—the model tended to favor the first image**. To mitigate this, we transitioned to a scoring-based method, designing detailed prompts to guide the evaluation process. This shift significantly improved the stability and accuracy of the evaluation.
> - **Validating the effectiveness of the evaluation framework**: We compared scores generated by the multimodal model with human-assigned scores and demonstrated the framework’s effectiveness and robustness using Pearson correlation. This method provides a unified, reproducible benchmark for personalized evaluation, addressing a key gap in current research.

---

> > ### Comment · Reviewer_w3GE · 2024-11-24
> >
> > Thank you for your efforts on the rebuttal, and it has corrected some of my misunderstanding and addressed my concerns. After carefully reading all reviews and response from authors, I raise my rating from 5 to 6.

---

> > > ### Author Response · Authors · 2024-11-24
> > > **Thank you!**
> > >
> > > Dear Reviewer w3GE,
> > >
> > > Thank you for your acknowledgment! We are glad that you are satisfied and all concerns are resolved.
> > >
> > > Best,\
> > > Authors

---

> ### Author Response · Authors · 2024-11-22
> **Supplementary explanation regarding "figure out which technical route is superior and where to head?"**
>
> Reviewer RNsM also discusses this point in his proposed weakness2. We plan to add another section to reinforce this point as well. Dedicated analysis of model performance across different generation types would strengthen our paper's alignment with this core objective of identifying superior technical approaches. We will incorporate the following comprehensive analysis in our revision:
>
> - **Concept Preservation Performance**
>   - **Animal:** Both fine-tuning and encoder-based personalization methods demonstrate strong performance in preserving animal concepts, primarily due to:
>     - Consistent visual features within species from a human cognition perspective
>     - For example, corgis exhibit highly unified visual representations, explaining their frequent use as test cases across personalization methods
>   - **Human:** Encoder-based methods significantly outperform fine-tuning approaches in human concept preservation, attributable to:
>     - Human observers' heightened sensitivity to facial feature accuracy
>     - Encoder-based methods have more precise capture and injection of subtle facial features
>     - Fine-tuning methods' tendency to focus on global image features rather than specific facial details
>     - This also confirms that current Identity-Preserving Generation (like InstantID) prefer encoder-based methods
>   - **Object:** The two approaches show complementary strengths
>     - Fine-tuning methods:
>       - Higher performance ceiling for common objects
>       - Better handling of objects with standardized appearances
>     - Encoder-based methods:
>       - Provide reliable baseline performance for rare/complex objects
>       - More consistent across diverse object types
>   - **Style:** Both approaches perform well in style preservation, likely because:
>     - Style features are more subjective and flexible
>     - Style transfer requires capturing high-level visual characteristics
>     - Both methods effectively learn and reproduce stylistic elements
> - **Prompt Following Capability**
>   - **Base Model Impact:** SDXL demonstrates superior prompt following compared to SD, driven by:
>     - More sophisticated and bigger text encoder architecture.
>     - Exposure to broader, more diverse training datasets and enhanced capacity for complex text-image relationships.
>   - **Methodological Differences:**
>     - Fine-tuning methods show clear advantages in prompt adherence due to:
>       - Preservation of original text-image conditional distribution
>       - Direct learning from image-text pairs
>       - Better maintenance of semantic alignment when properly regularized
>     - Encoder-based methods face challenges in prompt following due to:
>       - Potential disruption of learned text-image relationships during feature injection

---

### Public Comment · ~Liang_Chen10 · 2024-12-03
**Good benchmark!**

I think the authors provide a good benchmark targeting at evaluating a challanging and useful task (personalized image generation). The open source materials are helpful for others to follow and reproduce different baselines. Hoping to see larger scale dataset that might help creating better image generation model.

---

> ### Author Response · Authors · 2024-12-03
> **Thank you!**
>
> Dear Liang,
>
> Thank you for your comment! We appreciate your recognition of our work and focus on the challenging task of personalized image generation. We are committed to continuously pushing the scale and diversity of the benchmark.
>
> Best,\
> Authors

---

### Meta-Review · Area_Chair_PzLU · 2024-12-20

**Metareview:**

The paper introduces DreamBench++, a new benchmark dataset and evaluation metrics for personalized image generation. The authors propose a GPT-based automated evaluation that better aligns with human judgments. The benchmark contains a diverse set of images and prompts, and the authors analyze seven different personalized image generation methods, showing that the proposed metrics correlate more closely with human ratings than commonly used DINO and CLIP.

Strength
* The contribution of a new benchmark dataset and accompanying evaluation metrics is widely appreciated.
* The paper provides abundant analysis and experiments, showing valuable insights into the performance of various personalized image generation methods.

Weakness
* The necessity of the new benchmark compared to existing ones is not convincingly justified.
* The claimed diversity of the dataset and the generalizability of the GPT-based evaluation method are not sufficiently supported. Reviewers questioned potential biases in data source and the lack of evidence for the method’s transferability across different models.
* The authors claim that a single reference image is sufficient, but this is not well justified.
* The evaluation metrics do not cover all crucial aspects, such as overfitting.
* While the paper compares its approach mainly to DINO and CLIP, it lacks thorough comparisons with more recent or metrics (e.g., TIFA), and promised comparisons were not presented in the rebuttal.
* The reliance on GPT is not fully justified, and the scale of human annotations may be too limited for drawing robust conclusions.

Despite the concerns raised, the reviewers acknowledge the value of the new dataset and automated evaluation metrics presented in the paper. However, the contribution would be stronger if the authors: (1) provide a robust justification for the generalizability of the proposed metrics across various dimensions, (2) justify the design choice of using a single reference image or extend the benchmark to incorporate multiple references, (3) ensure the coverage of the proposed metrics, and (4) demonstrate that the scale and quality of human annotations are sufficient for reliable conclusions. Nevertheless, the paper is considered to meet the acceptance criteria if the authors incorporate the clarifications and additional materials provided in the rebuttal.

**Additional Comments On Reviewer Discussion:**

Overall, the reviewers found the rebuttal satisfactory, with the reviewers either maintaining the acceptance ratings or raising their ratings. Concerns about the necessity of a new benchmark, the dataset's diversity, the reliance on GPT, and the lack of comparisons with existing metrics were adequately addressed in the rebuttal. While some questions remain about the design of the benchmark dataset and evaluation metrics, the reviewers agree that these issues do not outweigh the merits of the benchmark and the overall contribution of the paper.

---

### Decision · Program_Chairs · 2025-01-22

Accept (Poster)